# Critical requirement of SOS1 for tumor development and microenvironment modulation in KRAS^G12D-driven lung adenocarcinoma

Fernando C. Baltanás [1,2] ✉, Rósula García-Navas[1], Pablo Rodríguez-Ramos [1], Nuria Calzada[1], Cristina Cuesta[3], Javier Borrajo[4], Rocío Fuentes-Mateos [1], Andrea Olarte-San Juan[1], Nerea Vidaña [1], Esther Castellano [3] & Eugenio Santos [1] ✉

The impact of genetic ablation of SOS1 or SOS2 is evaluated in a murine model of KRAS^G12D-driven lung adenocarcinoma (LUAD). SOS2 ablation shows some protection during early stages but only SOS1 ablation causes significant, specific long term increase of survival/lifespan of the KRAS^G12D mice associated to markedly reduced tumor burden and reduced populations of cancer-associated fibroblasts, macrophages and T-lymphocytes in the lung tumor microenvironment (TME). SOS1 ablation also causes specific shrinkage and regression of LUAD tumoral masses and components of the TME in pre-established KRAS^G12D LUAD tumors. The critical requirement of SOS1 for KRAS^G12D-driven LUAD is further confirmed by means of intravenous tail injection of KRAS^G12D tumor cells into SOS1^KO/KRAS^WT mice, or of SOS1-less, KRAS^G12D tumor cells into wildtype mice. In silico analyses of human lung cancer databases support also the dominant role of SOS1 regarding tumor development and survival in LUAD patients. Our data indicate that SOS1 is critically required for development of KRAS^G12D-driven LUAD and confirm the validity of this RAS-GEF activator as an actionable therapeutic target in KRAS mutant LUAD.

The Epidermal Growth Factor Receptor (EGFR) and KRAS, two pivotal components of RTK-ERK signaling pathways, are the most frequent targets for oncogenic driver mutations in lung adenocarcinoma (LUAD). Various tyrosine kinase inhibitor drugs are routinely used in the clinic against EGRF-driven LUAD tumors[1] but the therapeutic options against mutant KRAS-driven LUAD are much more limited.

Indeed, the first direct KRAS inhibitors have only been very recently approved for clinical testing[2–4] and the rapid appearance of resistance is an important concern for most new anti-KRAS drugs[5–8], a reason why the search must go on for new therapy approaches capable of inhibiting not only the function of RAS proteins but also that of upstream or downstream components in RTK-RAS signaling pathways[9–11]. The

[1]Lab 1. Cancer Research Center, Institute of Cancer Molecular and Cellular Biology, CSIC-University of Salamanca and CIBERONC, 37007 Salamanca, Spain. [2]Institute of Biomedicine of Seville (IBiS)/"Virgen del Rocío" University Hospital/CSIC/University of Seville and Department of Medical Physiology and Biophysics, University of Seville, Seville, Spain. [3]Lab 5. Cancer Research Center, Institute of Cancer Molecular and Cellular Biology, CSIC-University of Salamanca, 37007 Salamanca, Spain. [4]Departament of Biomedical Sciences and Diagnostic, University of Salamanca, 37007 Salamanca, Spain. ✉e-mail: fcalvo@us.es; esantos@usal.es

quest for new therapies must also take into account the extensive body of experimental data demonstrating that non-mutated (WT) cellular RAS proteins play also critical roles in the development of oncogenic RAS-driven tumors[12–17]. These observations, together with the emerging evidence indicating that different KRAS mutants may also exhibit varying sensitivities to GAP- or GEF-mediated modulation[18,19], support the notion that targeting other upstream regulators of RAS signaling pathways (such as SHP2 or the SOS family of RAS-GEF activators)[4,10,11,20,21] may also be a clinically beneficial approach.

SOS1 and SOS2 constitute the most universal and functionally relevant family of RAS-GEFs capable of activating RAS GTPases and their downstream signaling cascades in mammalian cells[9,22]. Despite their remarkably similar protein structures and expression patterns, SOS1 is essential for embryonic development[23,24] whereas SOS2 is dispensable to reach adulthood in mice[25]. Using a floxed, tamoxifen-inducible SOS1-null mutant, we managed to bypass the embryonic lethality of SOS1 KO alleles and to generate and analyze adult mice of 4 relevant SOS genotypes (SOS1/2$^{WT}$, SOS1$^{KO}$, SOS2$^{KO}$, and SOS1/2$^{DKO}$) thus making it possible to ascertain the functional specificity/redundancy of SOS1 and/or SOS2 in a variety of biological settings[26]. Under physiological conditions, we have demonstrated the functional prevalence of SOS1 over SOS2 regarding the control of fibroblast cellular proliferation and viability as well as a direct mechanistic link between SOS1 and control of intracellular redox status and mitochondrial homeostasis[27,28]. We and others have also identified specific functional roles of SOS1 and SOS2 in different cellular contexts[9,22,26,29–31]. Regarding tumoral contexts, we have also uncovered a critical functional contribution of SOS1 for development of DMBA/TPA-induced skin carcinogenesis[32] and BCR/ABL-driven chronic myeloid leukemia[33,34], thus supporting the consideration of this GEF isoform as a potentially useful therapeutic target for RAS-dependent cancers. In addition, SOS1 knockdown has also been shown to effectively enhance the therapeutic efficacy of MEK inhibitors in KRAS-amplified gastro-esophageal cancer[35] and to overcome secondary acquired resistance to osimertinib in EGFR-mutated LUAD human cell lines[36].

Regarding the consideration of SOS GEFs as molecular targets for tumor therapy it is also relevant to mention that, although mutant (oncogenic) RAS proteins are constitutively activated (GTP loaded) and, in theory, would not need the action of external GEF activators, various reports have shown that the cross-activation of WT RAS (which is SOS-dependent) by oncogenically mutated RAS is of critical importance for tumorigenic development[14–16]. Altogether, these reports and our observations of remarkable similarities between SOS-less and RAS-less cells[27,28,37] prompted us to postulate that SOS proteins may constitute valuable therapeutic targets in RTK-RAS-dependent cancers such as LUAD.

The tumor microenvironment (TME) is a capital player contributing to the development of lung tumors in vivo. In particular, different stromal cell populations including cancer-associated fibroblasts (CAFs) and tumor-associated macrophages (TAMs) are able to significantly impact tumor progression and impair tumor prognosis by modulating, via different mechanisms, the composition of the extracellular matrix as well as processes of angiogenesis, cellular migration or immune evasion in the tumors[38–40]. Interestingly, previous reports have also shown that SOS1 disruption markedly impairs proliferation and migration of MEFs[27,28,32], and also alters the ability of macrophages and neutrophils to be activated and migrate[32,41,42], further supporting the study of SOS1/2 GEFs as potential therapy targets for KRAS-mutated LUAD.

To extend and confirm the notion of SOS RAS-GEFs as critical functional requirements and actionable therapy targets in RAS-driven tumors, in this report we performed a detailed evaluation of the in vivo functional relevance and biological impact produced by genetically-mediated ablation of the SOS1 or SOS2 RAS-GEFs in a mouse model of KRAS$^{G12D}$-driven LUAD[43]. Our experimental observations demonstrate that SOS1 disruption specifically produces very significant therapeutic benefits, not only by markedly blocking development of the LUAD tumor cells but also by impairing the pro-tumorigenic effect of the surrounding stromal cells in the TME, thus supporting the consideration of SOS1 as a bona fide, actionable therapy target for developments of drugs/therapies against LUAD.

## Results

### SOS1 deficiency specifically protects from death and tumor-related pathophysiological defects in murine KRAS$^{G12D}$-driven LUAD

To allow for a direct experimental testing of the in vivo impact of ablation of the SOS1 or SOS2 RAS-GEFs, on the development of LUAD in mice, we cross-mated the KRAS$^{LA2}$ strain spontaneously developing KRAS$^{G12D}$-driven LUAD[43] with our previously described, TMX-inducible SOS1/2 KO system which allows to produce individual or combined ablation of SOS1 and/or SOS2 in mice[26]. These crosses gave rise to different genotypic combinations which, upon appropriate TMX treatment (at 1 month of age for specific SOS1 ablation), allowed to monitor and compare initiation and evolution of KRAS$^{G12D}$-induced LUAD in cohorts of littermates of the relevant SOS genotypes including SOS1/2$^{WT}$, SOS1$^{KO}$, or SOS2$^{KO}$ mice (Fig. 1a). Similar tests in mice lacking both SOS1 and SOS2 are not possible because SOS1/2$^{DKO}$ mice die precipitously after about two weeks of concomitant ablation of these RAS-GEF isoforms[26].

Interestingly, our evaluation of the lifespan and Kaplan–Meier survival curves of TMX-treated mice of the three relevant genotypic compositions (SOS1/2$^{WT}$/KRAS$^{G12D}$, SOS1$^{KO}$/KRAS$^{G12D}$, and SOS2$^{KO}$/KRAS$^{G12D}$) revealed that the single, specific ablation of SOS1, but not of SOS2, produced a significant improvement and extension of the lifespan and survival of KRAS$^{G12D}$-mutant mice as compared to their SOS1/2$^{WT}$/KRAS$^{G12D}$ control counterparts expressing normal levels of the two SOS RAS-GEFs (Fig. 1b).

We also evaluated various lung-related pathophysiological parameters in 12-month-old mice from these three KRAS$^{G12D}$ experimental groups in comparison to a similarly aged cohort of healthy SOS1/2$^{WT}$/KRAS$^{WT}$ animals used as controls of the normal, physiological value for each of those parameters (Fig. 1c; Supplementary Fig. S1). As expected, the SOS1/2$^{WT}$/KRAS$^{G12D}$ mice showed a significant deterioration in comparison to healthy SOS1/2$^{WT}$/KRAS$^{WT}$ controls with regards to percentage O$_2$ saturation in blood (Fig. 1c) as well as respiratory rate (Supplementary Fig. S1a, b), or electrocardiogram recording of cardiac activity and frequency (Supplementary Fig. S1c). However, consistently with the significantly improved survival data (Fig. 1b), SOS1 ablation also caused a specific rescue of all those disease-dependent pathophysiological parameters. Indeed, the SOS1$^{KO}$/KRAS$^{G12D}$ mice showed similar physiological profiles and values to those of healthy SOS1/2$^{WT}$/KRAS$^{WT}$ controls, in clear contrast with the impaired parameters displayed by the SOS1/2$^{WT}$/KRAS$^{G12D}$ and SOS2$^{KO}$/KRAS$^{G12D}$ mice (Fig. 1c; Supplementary Fig. S1).

### SOS1/2 ablation markedly impairs initiation and subsequent malignant progression of KRAS$^{G12D}$-driven LUAD lung tumors

To evaluate the role of SOS1 and SOS2 during the early phases of KRAS$^{G12D}$-driven LUAD tumorigenesis, cohorts of young SOS1/2$^{WT}$/KRAS$^{G12D}$, SOS1$^{fl/fl}$/KRAS$^{G12D}$, and SOS2$^{KO}$/KRAS$^{G12D}$ mice were equally fed after 1 month of age with TMX-containing diet (to induce SOS1 disruption), and the development of LUAD tumors was then monitored in detail during the next ensuing months (Fig. 1d–i). Our quantification of the number of tumors visible on the pleural surface of the lungs of 1-month-old (TMX-untreated) mice showed initial tumor count that appeared to be already slightly reduced in the SOS2$^{KO}$/KRAS$^{G12D}$ group as compared to their SOS1/2$^{WT}$/KRAS$^{G12D}$ and SOS1$^{fl/fl}$/KRAS$^{G12D}$ counterparts (Fig. 1d). In addition, after starting the TMX treatment, the

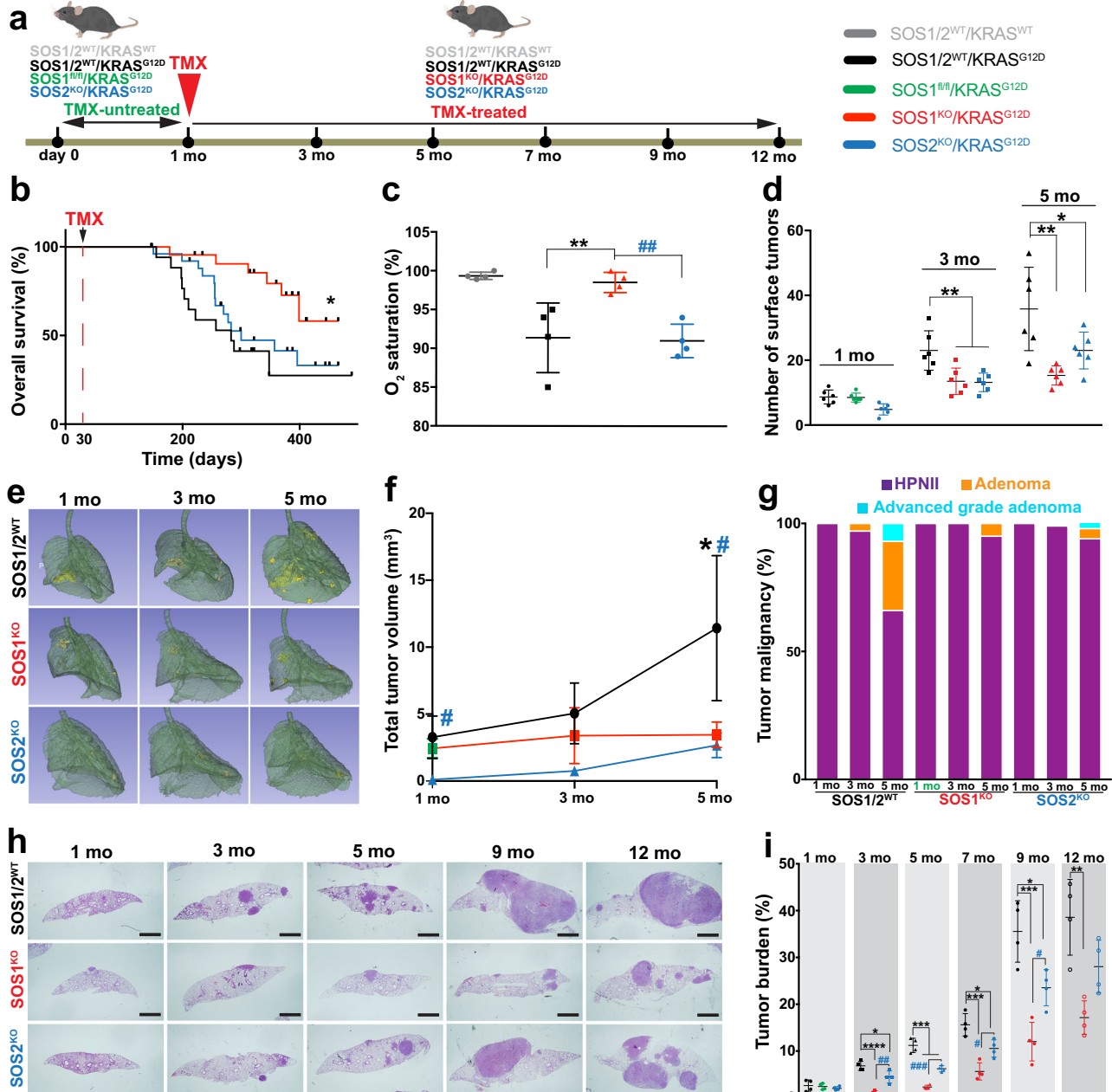

**Fig. 1 | SOS1/2 genetic ablation protects from KRAS^G12D-driven LUAD in mice.**
**a** Schematic illustration of the experimental strategy applied to all different genotypic groups. Animals of the indicated KRAS and SOS1/2 genotypes were equally treated with TMX after 1 month of age and then analyzed during the ensuing months at the timepoints indicated in the following panels. **b** Kaplan–Meier survival plots for SOS1/2^WT (black), SOS1^KO (red) and SOS2^KO (blue) animals analyzed in our model of RAS^G12D-driven LUAD. $n = 20$ (SOS1/2^WT), $n = 25$ (SOS1^KO), $n = 28$ (SOS2^KO); $*P = 0.0107$ vs SOS1/2^WT. Log-rank (Mantel-Cox) test. **c** Percentage of $O_2$ saturation in blood of 12-month-old SOS1/2^WT/KRAS^WT and KRAS^G12D mice of the SOS1/2 genotypes. $n = 4$ independent mice per genotype. $**P = 0.0097$ vs SOS1/2^WT/KRAS^G12D; $^{\#\#}P = 0.0068$ vs SOS2^KO. Data shown as mean ± SD. One-way ANOVA and Tukey's test. **d** Total number of KRAS^G12D-mutant tumors present on the surface of the lungs of 1, 3, and 5 months-old SOS1/2^WT, SOS1^KO, or SOS2^KO mice. $n = 6$ independent mice per genotype. Data shown as mean ± SD. At 3 months: $**P = 0.0067$ vs SOS1^KO and $**P = 0.0052$ vs SOS2^KO; at 5 months: $**P = 0.0018$ vs SOS1^KO and $*P = 0.0424$ vs SOS2^KO). One-way ANOVA and Tukey's test. **e** Representative lung microCT scan images for each age and genotype. **f** Progression of LUAD tumor volume in the groups and timepoints indicated. $n = 3$ independent mice per genotype. Data

expressed as mean ± SD. At 1 month: $^{\#}P = 0.0201$ vs SOS1^KO and $P = 0.0052$ vs SOS2^KO; at 5 months: $*P = 0.0382$ vs SOS1^KO and $^{\#}P = 0.0264$ vs SOS2^KO). One-way ANOVA and Tukey's test. **g** Percentage distribution of the histopathological grades exhibited by KRAS^G12D lung tumors of the indicated genotypes and timepoints. Hyperproliferating Type II Pneumocytes (HPNII, black bars), adenoma (orange bars) or advanced grade adenoma (light blue bars). **h, i** KRAS^G12D-driven tumor progression examined in lungs of 1 to 12-month-old KRAS^G12D-mutant mice of the indicated genotypes. **h** Representative H&E-stained sections of lungs from animals of the indicated age and genotype. Scale bars, 1 mm. **i** Kinetics of tumor burden progression from 1-month-old to 12-month-old mice of the indicated genotypes. $n = 4$ independent mice per genotype and timepoint. Data shown as mean ± SD. At 3 months: $*P = 0.0176$ vs SOS2^KO; $****P < 0.0001$ vs SOS1^KO and $^{\#\#}P = 0.0019$ vs SOS2^KO; at 5 months: $***P \leq 0.001$ vs SOS1^KO and SOS2^KO and $^{\#\#\#}P = 0.0008$ vs SOS2^KO; at 7 months: $***P = 0.0002$ vs SOS1^KO and $*P = 0.0164$ vs SOS2^KO and $^{\#}P = 0.0195$ vs SOS2^KO; at 9 months: $***P = 0.0002$ vs SOS1^KO and $*P = 0.0195$ vs SOS2^KO and $^{\#}P = 0.0243$ vs SOS2^KO; at 12 months: $**P = 0.0019$ vs SOS1^KO. One-way ANOVA and Tukey's test. Source data are provided as a Source data file.

total tumor count in 3-month-old and 5-month-old TMX-treated SOS1[KO]/KRAS[G12D] and SOS2[KO]/KRAS[G12D] mice was dramatically reduced in comparison to their control SOS1/2[WT]/KRAS[G12D] counterparts, indicating a striking dependence on SOS1/2 during the early stages of KRAS[G12D]-driven LUAD initiation and development (Fig. 1d). Consistent with this, analysis of in vivo microCT scanning images of young animals of the different genotypes during the same time period confirmed that the volume of the tumors in the lungs of these young mice was very markedly reduced in the two SOS1[KO] and SOS2[KO] strains as compared to their SOS1/2[WT]/KRAS[G12D] counterparts during these early stages of LUAD development (Fig. 1e, f). We also evaluated the histopathological grade reached by the KRAS[G12D]-driven lung tumors in each experimental group at specific timepoints of analysis (Fig. 1g). At 1 month of age (when TMX-induced SOS1 ablation in SOS1[fl/fl]/KRAS[G12D] mice was just started), practically 100% of the KRAS[G12D]-driven lung tumors in all the experimental groups corresponded to benign, hyperproliferating type II pneumocytes (HPNII) (Fig. 1*g*). As expected, during the ensuing months the percentage of benign adenomas increased steadily in the control SOS1/2[WT]/KRAS[G12D] mice, with 3% at 3 months of age, 30% at 5 months of age, and about 7% of more advanced grade adenomas already detectable in 5-months-old animals (Fig. 1g). In sharp contrast, the vast majority of the KRAS[G12D]-driven lung tumors found in SOS1[KO] or SOS2[KO] mice were characterized as benign HPNII (~95% at 5 months), and a slower tumor grade progression was also observed in SOS1[KO] animals as compared to SOS2[KO] mice, with advanced grade adenomas constituting ~3% of the tumors present in 5-month-old SOS2[KO] mice, whereas they were completely absent in similarly aged SOS1[KO] counterparts (Fig. 1g).

Long-term morphological examination of fixed tissue sections of the lungs of mice of the relevant SOS1/2 genotypes allowed a detailed quantitative evaluation of the impact of SOS1 or SOS2 ablation on the development and evolution of KRAS[G12D] lung tumors (Fig. 1h, i). Consistent with our prior observations, quantitative analysis of H&E-stained sections of lungs extracted at different timepoints during tumor progression confirmed that tumor burden was significantly reduced in the lungs of SOS1/2[KO]/KRAS[G12D] mice as compared to those of their SOS1/2[WT]/KRAS[G12D] counterparts during the early stages of tumorigenesis (Fig. 1h, i). Importantly, our long-term analyses until 12 months of age (when most diseased animals had to be euthanized according to ethical regulations for animal experimentation) demonstrated in particular that the LUAD tumor burden was always significantly reduced in SOS1[KO]/KRAS[G12D] mice in comparison to SOS2[KO]/KRAS[G12D] mice for the whole duration of the life of these animals (Fig. 1h, i), clearly indicating a preferential functional contribution of SOS1 over SOS2 with regards to both the early stages of initiation and the later stages of progression of KRAS[G12D]-driven LUAD in mice.

The prevalent role of SOS1 over SOS2, in KRAS[G12D]-driven LUAD development in mice was also consistent with the significant reduction of cellular proliferation (Ki67-positive tumor cells, Fig. 2a) and ERK activation (pERK levels, Fig. 2b) observed by IHC in lung tumors from 5-months-old SOS1[KO] mice as compared to similarly aged SOS1/2[WT] or SOS2[KO] mice. Our detailed molecular assays confirmed the silencing of SOS1 and SOS2 in the lung tumors isolated from SOS1[KO] and SOS2[KO] mice and also showed that SOS1 or SOS2 depletion correlated with significant reduction of RAS-GTP levels in those tumors (Fig. 2c, d).

## SOS1 ablation specifically hinders the TME of KRAS[G12D]-driven LUAD in mice

We also investigated whether SOS1 or SOS2 ablation may affect the presence or the response of fibroblasts, macrophages or lymphocytes in the stromal TME of KRAS[G12D]-driven LUAD in mice (Fig. 3). Using smooth muscle actin (SMA) immunostaining as a specific marker of cancer-associated fibroblasts (CAFs)[44] in the TME of the lung tumors, we observed that SOS1 (but not SOS2) ablation dramatically reduced detection of SMA-positive CAFs in the TME of the lung tumors of 5-

month-old KRAS[G12D]-mice as compared to their SOS1/2[WT] and SOS2[KO] counterparts (Fig. 3a). Consistent with this, Masson's trichrome staining detecting CAF-dependent collagen deposition[44] in the TME was also very strongly reduced in the lungs of SOS1-ablated mice as compared to the SOS1/2[WT] and SOS2[KO] experimental groups (Fig. 3b). These observations suggest a very relevant functional role of SOS1 regarding recruitment and activity of CAFs in the TME of KRAS[G12D]-driven LUAD, whereas SOS2 appears to be rather dispensable for these pathophysiological processes.

The impact of SOS1 or SOS2 ablation on immunity-related cell populations present in the TME of KRAS[G12D]-driven lung tumors was also examined. Thus, CD68 immunostaining identifying TAMs in the TME[45] showed that SOS1 (but not SOS2) ablation significantly reduced the presence of these immunopositive macrophages within the TME of the tumors (Fig. 3c). Furthermore, consistent with our prior reports indicating that SOS1 removal severely affected normal T lymphocyte maturation[26], CD3-immunostaining[29] demonstrated also that SOS1 ablation strongly reduced T lymphocytic infiltration in KRAS[G12D]-driven lung tumors, in contrast to the tumors of SOS1/2[WT] or SOS2[KO] counterparts (Fig. 3d).

The functional involvement of SOS1 in regulation of the immunological environment in the TME of KRAS[G12D]-driven LUAD was further confirmed by measuring the relative levels of various chemokines and cytokines present in the TME of the lung tumors of our three experimental groups (Supplementary Fig. S2). Among other changes, our data documented a statistically significant reduction in levels of pro-inflammatory signals such as IL-1, IL1-r, or MIP1 when SOS1 is absent (Supplementary Fig. S2). On the other hand, CD31 immunostaining was used to evaluate whether SOS1 or SOS2 depletion altered the vasculature[32] of KRAS[G12D]-driven lung tumors and showed that neither SOS1 nor SOS2 ablation impacted significantly the amount of vasculature in KRAS[G12D]-driven LUAD when comparing the tumors arising in SOS1/2[WT] mice to those of SOS1[KO] or SOS2[KO] strains (Fig. 3e).

Overall, these observations support the notion that SOS1 disruption also markedly impacts different cell populations involved in deleterious effects mediated by the TME in KRAS[G12D]-driven LUAD, suggesting an additional, potential therapeutic benefit of SOS1 removal with regard to KRAS-mutated lung tumors.

## SOS1 ablation causes tumor regression in preexisting KRAS[G12D]-driven lung tumors

After uncovering the crucial contribution of SOS1 to KRAS[G12D]-driven LUAD regarding initiation and progression of both the lung tumoral cells and their surrounding TME, we wished to determine whether SOS1 ablation could also have an impact on preexisting SOS1/2[WT]/KRAS[G12D] lung tumors by impairing maintenance/progression, or even inducing regression, of those tumors. To this end, we compared a series of relevant tumor parameters between lung tumor samples extracted from untreated, 6-month-old, SOS1[fl/fl]/KRAS[G12D] mice and tumor samples of 6-month-old animals of the same genotype after undergoing 2 months of TMX treatment to actually become SOS1[KO]/KRAS[G12D] (Fig. 4). Strikingly, the comparison between microCT scans (Fig. 4a) or H&E-stained sections (Fig. 4b) of those tumors uncovered a very significant shrinkage of the volume of the pre-existing SOS1[WT] (TMX-untreated, SOS1[fl/fl]) tumors in comparison to their subsequently TMX-treated (SOS1[KO]) counterparts.

Consistent with our prior observations during the early stages of tumor initiation/progression, the tumor shrinkage produced by TMX-induced SOS1 ablation in preexisting, fully developed, tumors was also mechanistically associated to a significant reduction of cell proliferation (Ki67, Fig. 4c) and ERK phosphorylation (Fig. 4d) in the SOS1-deficient tumors. Likewise, SOS1 ablation was also associated with a significant decrease of immunoreactivity against SMA (CAFs, Fig. 4e) and CD68 (TAMs, Fig. 4f) detected after ablation of SOS1 in the

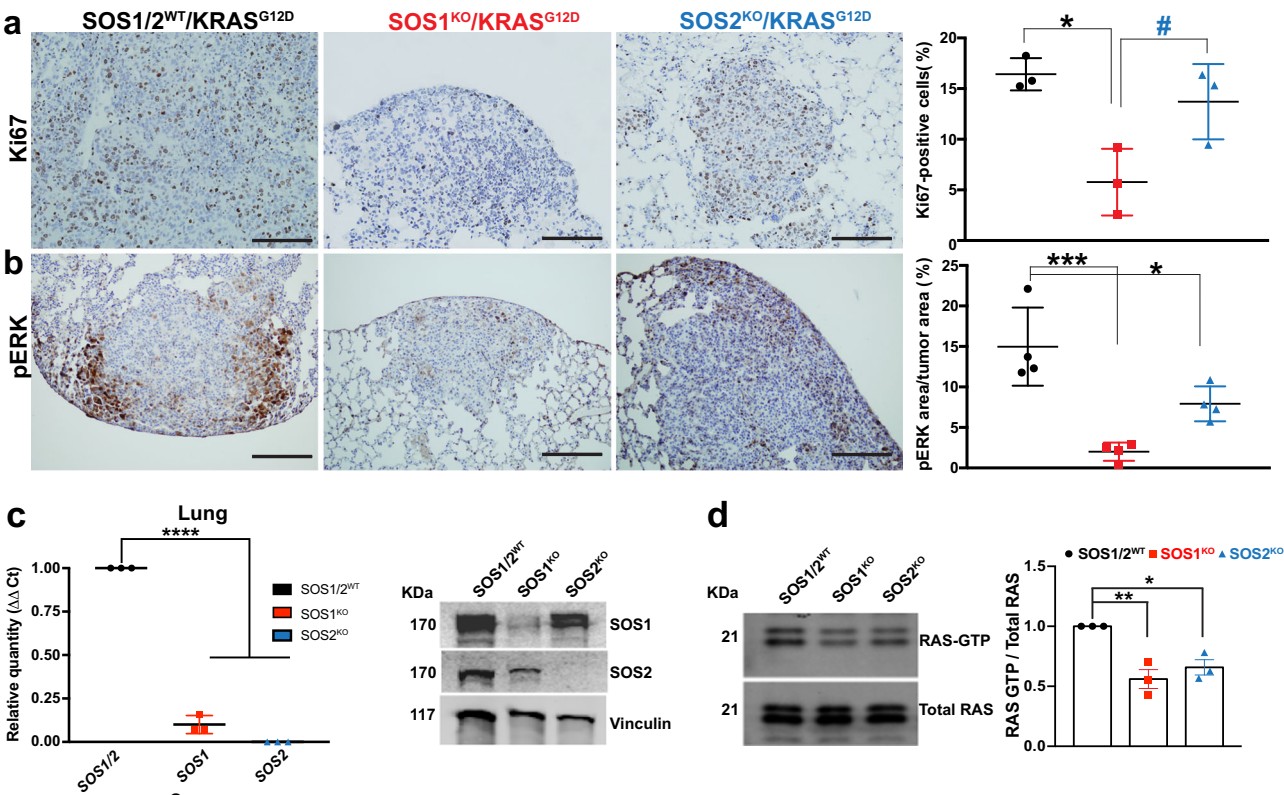

**Fig. 2 | SOS1 depletion reduces cell proliferation and ERK phosphorylation rates in KRAS$^{G12D}$ lung tumors. a, b** Representative images of paraffin-embedded sections from lungs of 5-month-old, KRAS$^{G12D}$-mutant mice of the indicated SOS genotypes (SOS1/2$^{WT}$, SOS1$^{KO}$, and SOS2$^{KO}$) after immunostaining against Ki67 (**a**) and pERK (**b**). The bar charts on the right side quantitate the percentage of Ki67-positive cells relative to total number of cells in the tumors (**a**) and the percentage of pERK-stained area in the tumor relative to total tumor area (**b**) in the samples. Scale bars, 100 μm. $n = 3$ independent mice per genotype in (**a**) and $n = 4$ independent mice per genotype in (**b**). Data expressed as mean ± SD; * vs SOS1/2$^{WT}$; # vs SOS1$^{KO}$. In (**a**) (*$P = 0.0117$ and #$P = 0.0412$) and in (**b**) (***$P = 0.0006$ and *$P = 0.0266$). One-way ANOVA and Tukey's test. **c** SOS1 and SOS2 expression levels in lung tumors isolated from TMX-treated animals of the relevant SOS genotypes (SOS1/2$^{WT}$, SOS1$^{KO}$, SOS2$^{KO}$). Left: mRNA quantification by means of RT-qPCR analysis using β-2-microglobulin as internal control for normalization. Right: Representative Western blots of SOS1 and SOS2 protein expression using vinculin as internal control for normalization. $n = 3$ independent samples per genotype. Data shown as mean ± SD. ****$P < 0.0001$ vs SOS1/2$^{WT}$. One-way ANOVA and Tukey's test. **d** Representative RAS-GTP assays and corresponding quantitative densitometric analyses performed in KRAS$^{G12D}$-lung tumors of 5-month-old, TMX-treated SOS1/2$^{WT}$ and SOS1/2-deficient mice. $n = 3$ independent samples per genotype. Data shown as mean ± SD. *$P = 0.0146$ and **$P = 0.0044$ vs SOS1/2$^{WT}$. One-way ANOVA and Tukey's test. Source data are provided as a Source data file.

---

preexisting (SOS1$^{WT}$) SOS1$^{fl/fl}$ lung tumors. On the other hand, similar, almost negligible rates of cell death (CC3, Fig. 4g) were observed in both SOS1$^{WT}$ and SOS1$^{KO}$ tumors, suggesting that SOS1 does not play a significant role priming apoptosis in KRAS$^{G12D}$-driven lung tumor cells.

## Ablation of SOS1 specifically impairs nesting and tumor progression of KPB6 tumor cells in an orthotopic model of lung tumorigenesis

The above observations showing the remarkable impact caused by SOS1 ablation on tumor initiation, progression and stromal microenvironment (Figs. 1–3) were generated in an experimental system involving wide-body elimination of SOS1 or SOS2 in our KRAS$^{G12D}$ mouse model and, therefore, we could not discern whether those clinically beneficial effects are cell-autonomous or not. To address this question, we used here an orthotopic model of LUAD tumorigenesis using KPB6 mouse LUAD cells harboring KRAS$^{G12D}$ and p53 mutations[46] where we analyzed the role of SOS1 and SOS2 in tumor cell homing and lung colonization after eliminating these RAS-GEFs in either the recipient mice or in the injected KPB6 tumor cells.

We first assessed the effect of the absence of SOS1 or SOS2 in the pulmonary tissues of the recipient animals by evaluating lung tumor development and progression after intravenous tail injection of tumoral KPB6 LUAD cells into syngeneic SOS1/2$^{WT}$/KRAS$^{WT}$, SOS1$^{KO}$/KRAS$^{WT}$ or SOS2$^{KO}$/KRAS$^{WT}$ mice (Fig. 5a). Strikingly, our results showed

that the absence of SOS1, but not SOS2, in the lungs of the recipient, injected mice dramatically reduced the tumor burden of KPB6-dependent lung tumors as compared to their SOS1/2$^{WT}$/KRAS$^{WT}$ WT counterparts (Fig. 5b, c). Interestingly, the rates of tumor cell proliferation (Fig. 5d) and pERK phosphorylation (Fig. 5e) levels measured within the lung tumors were not significantly affected by either SOS1 or SOS2 ablation in comparison to their SOS1/2$^{WT}$/KRAS$^{WT}$ WT controls (Fig. 5d, e), although the low levels of Ki67 expression detectable in non-tumor areas of the lung appeared to be slightly reduced in the SOS1$^{KO}$ samples. Furthermore, the presence of CAFs in the TME of KPB6-driven lung tumors was also specifically and significantly reduced in SOS1-depleted SOS1$^{KO}$/KRAS$^{WT}$ mice as compared to both their WT (SOS1/2$^{WT}$/KRAS$^{WT}$) and SOS2$^{KO}$ (SOS2$^{KO}$/KRAS$^{WT}$) counterparts (Fig. 5f).

Conversely, we also evaluated the effect of the specific removal of SOS1 or SOS2 in the injected KPB6 cells whereas the pulmonary tissues of the recipient mice expressed normal levels of SOS1 and SOS2 (Fig. 6a). For this purpose, KPB6 cells devoid of SOS1 or SOS2 by means of CRISPR/Cas9 (Fig. 7a) were intravenously injected into WT recipient mice SOS1/2$^{WT}$/KRAS$^{WT}$ and, under these experimental conditions, our analyses showed that SOS1 ablation in KPB6 cells resulted in significantly reduced lung tumor burden (H&E staining, Fig. 6b, c) and cellular proliferation (Ki67, Fig. 6d) in comparison to the WT controls. Somewhat surprisingly, these studies showed that CRISPR/Cas9-

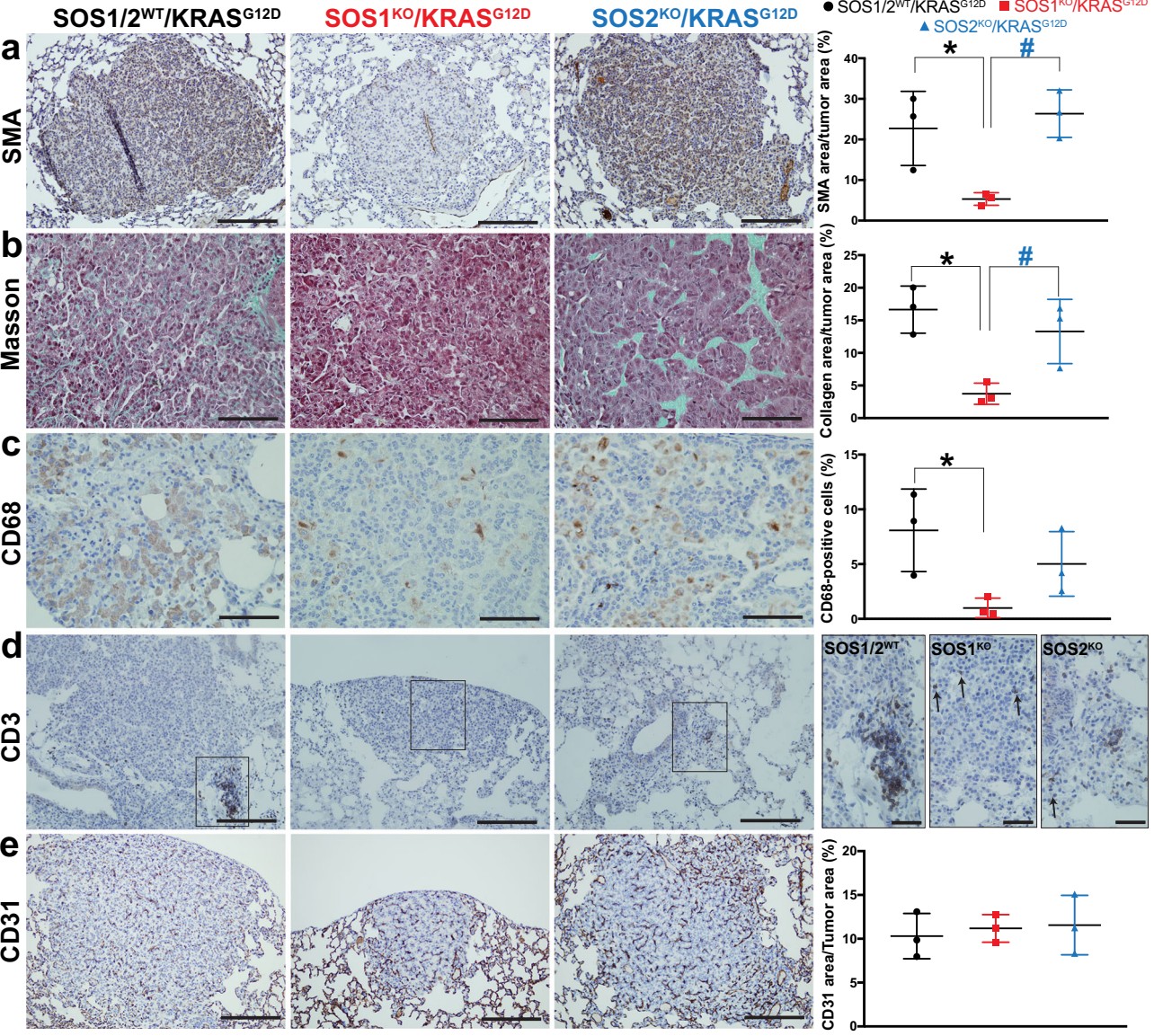

**Fig. 3 | SOS1 depletion modulates the tumor microenvironment in KRAS^G12D-driven LUAD. a–e** Representative images of paraffin-embedded sections from KRAS^G12D-driven lung tumors extracted from 5-month-old, SOS1/2^WT, SOS1^KO, and SOS2^KO mice after immunolabeling for SMA (**a**), CD68 (**c**), CD3 (**d**), and CD31 (**e**), or staining with Masson trichrome (**b**), as indicated. Scale bars: **a**, **d**, **e** 100 μm; **b**, **c** 50 μm; **d** inserts, 50 μm. Bar charts on the right side represent the percentage of total tumor area that was specifically stained for SMA (**a**), collagen (**b**), and CD31 (**e**) or the percentage of total tumor cells that were immunopositive for CD68 (**c**) in lungs of the indicated genotypes. $n = 3$ independent mice per genotype. Data shown as mean ± SD. *$P < 0.05$ vs SOS1/2^WT; #$P < 0.05$ vs SOS1^KO. In (**a**): *$P = 0.035$ and #$P = 0.0146$. In (**b**): *$P = 0.0107$ and #$P = 0.0404$. In (**c**): *$P = 0.0488$. One-way ANOVA and Tukey's test. Source data are provided as a Source data file.

mediated removal of SOS2 also caused some reduction of tumor burden and percentage of Ki67-positive and pERK-positive cells in the tumoral masses (Fig. 6b–e). In contrast, no significant differences were observed regarding the presence of CAFs in the TME of tumors driven by KPB6 cells devoid of SOS1 or SOS2 in comparison to their CRISPR/Cas9 control counterparts (Fig. 6f). Cell count measurements in cultures of CRISPR-modified KPB6 cells showed that both SOS1 or SOS2 ablation significantly reduced proliferative rate (Fig. 7b) and RAS-GTP levels (Fig. 7c) in comparison to CRISPR^Cas9 control KPB6 cells. Consistently, pharmacological treatment with BAY-293, a selective inhibitor of SOS1:KRAS interaction[47], resulted also in decreased cell proliferation of different KRAS^G12D-LUAD murine tumor cell lines such as KPB6, LKR10, and LKR13, exhibiting a closely similar anti-proliferative effect than the specific KRAS^G12D inhibitor MRTX1133 (Fig. 7d). However, no synergistic effects of BAY-293 and MRTX1133 were observed in those assays when both drugs were used in combination

on any of these murine cell lines (Fig. 7d). In this regard, reduced rates of RAS activation upon EGF stimulation were also detected in BAY-293-treated or MRTX1133-treated LUAD cell lines, although in this case the inhibitory effect of MRTX1133 appeared to be slightly higher than that produced by BAY-293 (Fig. 7e).

### In silico analysis reveals a dominant functional contribution of SOS1 to human KRAS-driven LUAD

The above experimental data in mice point to a strong, crucial functional involvement of SOS1 in the development of murine KRAS-driven LUAD. In addition, we have also been able to document here the unique dependence on SOS1 of human KRAS-driven LUAD by means of in silico analyses of publicly available datasets for human LUAD cancer cell lines and patients (Fig. 8).

We first evaluated the impact of *hSOS1* or *hSOS2* ablation in a wide battery of human LUAD cell lines by means of unsupervised

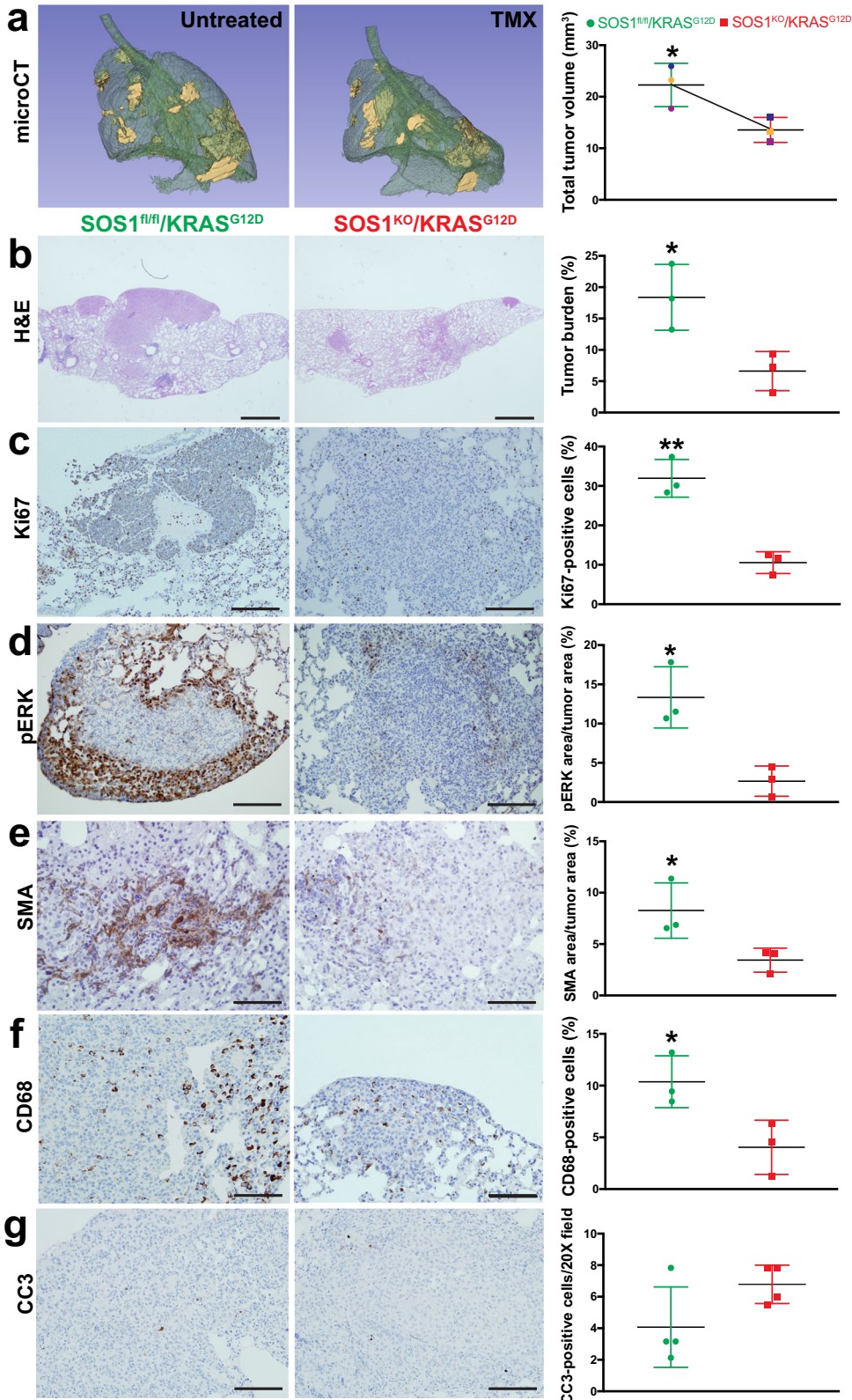

computational analysis of genomic data available in the CRISPR data-set of the Cancer Dependency Map portal[48]. Consistent with our previous observations in mice, our analysis of the subset of LUAD cell lines assigned a higher dependency score to *hSOS1* regarding regulation of cell proliferation, in comparison with *hSOS2*, which showed an almost null dependency score in this regard (Fig. 8a). Furthermore, our analysis of correlation between the expression levels of *hKRAS* and those

of *hSOS1* or *hSOS2* in the same subset of human LUAD cell lines yielded a direct, statistically significant correlation parameter between *hKRAS* and *hSOS1*, in contrast to the almost negligible correlation values calculated for *hSOS2* and *hKRAS* gene expression (Fig. 8b).

We also used the well-annotated datasets contained in the CAN-CERTOOL database[49] to evaluate the impact of *hSOS1* and *hSOS2* expression on human LUAD development and survival from this

**Fig. 4 | Impact of SOS1 ablation on preexisting KRAS$^{G12D}$-driven lung tumors. a** Representative microCT scanning image of the lungs of a TMX-untreated 4-month-old SOS1$^{fl/fl}$/KRAS$^{G12D}$ mouse (left) and the lungs of the same animal (right) after two additional months (6-month-old) of TMX treatment. The graph data points corresponding to each individual mouse are identified by a distinctive color in each case. **b–g** Representative images of paraffin-embedded sections from the lungs of 6-month-old mice (SOS1$^{fl/fl}$/KRAS$^{G12D}$, TMX-untreated, left side) and comparable counterpart lungs samples from 6-month-old mice treated with TMX for a two-months period (starting treatment at 4 months of age) that were stained with H&E (**b**) or immunostained for Ki67 (**c**), pERK (**d**), SMA (**e**), CD68 (**f**) and cleaved-caspase 3 CC3 (**g**) as indicated. Scale bars: **b**, 1 mm; **c, d, g** 100 μm; **e, f** 200 μm. *n* = 3 independent mice per genotype and experimental condition. The bar charts on the right side depict quantitative comparisons between the images on the left side pictures (green bars: lungs of TMX-untreated mice; red bars: lungs of TMX-treated mice for 2 months) and represent, respectively, the percentage of lung tumor volume (**a**), the percentage of lung tumor burden (**b**), the percentage of Ki67-positive cells in the tumor (**c**), the percentage of total lung tumor area that was immunostained for pERK (**d**) or for SMA (**e**), or the percentage of CD68-positive cells (**f**) and CC3-positive cells per 20X microscopy field (**g**). *n* = 3 independent mice per genotype and experimental condition. Data shown as mean ± SD. *$P$ < 0.05 vs SOS1$^{KO}$; **$P$ < 0.01 vs SOS1$^{KO}$. Two-tailed paired t-test was used for statistical analysis in panel **a** ($P$ = 0.0173), whereas two-tailed unpaired t-test was employed to analyze the comparisons in panels (**b–g**) (**b**: $P$ = 0.0289; **c**: $P$ = 0.0026; **d**: $P$ = 0.0132; **e**: $P$ = 0.0466; **f**: $P$ = 0.0389). Source data are provided as a Source data file.

disease (Fig. 8c, d). Comparison of *hSOS1* and *hSOS2* gene expression between non-tumoral (N) and tumoral (LUAD) human samples from the Okayama dataset[49] revealed that *hSOS1* (but not *hSOS2*) is significantly upregulated in LUAD tumor samples as compared to non-tumoral lung samples (Fig. 8c). Regarding patient survival, a quartile-based discrimination of patients carried out by CANCERTOOL[49] on the basis of *SOS1* or *SOS2* gene expression showed that LUAD patients with highest *SOS1* gene expression levels tend to exhibit the lowest overall survival rates, whereas *SOS2* expression level appeared not to affect the survival rates (Fig. 8d).

## Discussion

Here, we cross-mated a well-known murine model of KRAS$^{G12D}$-driven LUAD[43] with our SOS1/2-KO system[26–28] to generate resulting SOS1/2$^{WT}$, SOS1$^{KO}$, and SOS2$^{KO}$ mouse litters expressing KRAS$^{G12D}$ that allowed us to effectively evaluate the functional relevance/contribution of the RAS-GEF activators SOS1 and SOS2 to KRAS-driven LUAD processes. Overall, our analyses of the survival curves and the specific patho-physiological phenotypes displayed by our different mouse strains devoid of SOS1 or SOS2 clearly demonstrated a critical requirement of SOS1 for in vivo initiation, maintenance, and progression of KRAS$^{G12D}$-driven LUAD in mice. Interestingly, and consistent with previous reports[31,50,51] a detailed temporal analysis of our different KRAS$^{G12D}$ mouse strains uncovered also a measurable delaying effect of SOS2 ablation regarding the onset and initial stages of LUAD development in KRAS$^{G12D}$ mice. Nevertheless, our results clearly showed that the LUAD-blocking effect of SOS1 ablation was always significantly higher than that of SOS2 ablation, as only the SOS1-ablated mice displayed a sustained, significant reduction of lung tumor burden and pathological grade progression of the tumor masses and their surrounding TME throughout the life of these animals. Indeed, only SOS1$^{KO}$ mice, but not SOS2$^{KO}$ mice, showed a significant improvement of the long-term survival of tumor-bearing KRAS$^{G12D}$ mice. Furthermore, we also documented a specific, remarkable regression caused by SOS1 ablation on preexisting tumors of 6-month-old mice in comparison to similarly aged, tumor bearing, SOS1/2$^{WT}$ or SOS2$^{KO}$ mice.

All these observations uncover a significant, dominant functional contribution of SOS1 to LUAD development and support the notion that, as for other cellular functionalities previously characterized in single and double SOS1/2-KO strains[26–28,30,32], SOS2 may play partially overlapping or ancillary roles relative to SOS1 during early stages of LUAD initiation in KRAS$^{G12D}$ mice[22,26]. The critical role of SOS1 regarding LUAD development is further supported from the mechanistic point of view by the specific inhibition produced by SOS1 ablation, but not SOS2 ablation, on the rates of cellular proliferation (Ki67) and ERK activation (pERK) detected in the lung tumoral masses of SOS1$^{KO}$ mice as compared to their SOS1/2$^{WT}$ or SOS2$^{KO}$ counterparts. An essential functional contribution of SOS1 to LUAD is also consistent with the known landscape of genomic alterations in LUAD[52], which preferentially impact components of RTK/RAS/ERK signaling, where SOS1 is a key regulator of downstream signaling not only by normal, non-mutated RAS proteins but also by oncogenically mutated, cancer-

inducing RAS isoforms[9,52]. Our current demonstration of the critical mechanistic requirement of SOS1 for LUAD development is also consistent with prior reports demonstrating the critical requirement of SOS1 for development of KRAS$^{G12D}$-induced myeloproliferative neoplasms[17] as well as for BCR/ABL-driven leukemogenesis[33,34] and for chemically-induced skin carcinogenesis[32].

The TME is recognized to play critically relevant roles, not only in control of tumor progression itself but also in modulation of the clinical responses or resistances elicited by different cancer therapies[38,40]. Interestingly, the marked therapeutic effect of SOS1 ablation on LUAD development was not limited to a significant reduction of the tumoral masses, but also impacted significantly the pro-tumoral activity exerted by various cell subpopulations of the stromal LUAD microenvironment. Thus, we observed that SOS1 ablation, but not SOS2 ablation, was specifically linked to a significant reduction of the number of CAFs recruited to the tumoral masses and their collagen-deposition activity in them. Furthermore, SOS1 depletion was also specifically linked to a marked reduction of T-lymphocyte infiltration in the tumors as well as to significant reduction of various pro-inflammatory cytokines secreted by the tumors. These TME alterations are specific and not due to any overall change in the hematopoietic composition of the single SOS1$^{KO}$ mice analyzed here since our previous studies demonstrated that only the double knockout SOS1/SOS2$^{DKO}$ animals (but not the single SOS1$^{KO}$ or SOS2$^{KO}$ mice) display detectable changes of hematopoietic composition in comparison to WT mice[26]. On the other hand, no specific effect of SOS1 ablation was detected on the level of vasculature displayed by these lung tumors. Specifically, our detailed analyses of our KRAS$^{G12D}$ LUAD model clearly demonstrate that SOS1 ablation impacts both the homeostasis/proliferation of the intrinsic tumoral cell population as well as that of extrinsic cell populations such as the CAFs or TAMs of the TME. These observations in our LUAD mouse system are consistent with prior reports in different KO biological settings demonstrating the essential contribution of SOS1 to the maintenance/homeostasis of various cell types that may be eventually recruited to the TME, including MEFs[27,28], macrophages[32,41], neutrophils[42], or lymphocytes[26,29].

Besides its impact on the TME, it was also relevant that the effect of SOS1 ablation was manifested not only by inhibiting and delaying tumor development and burden during early LUAD stages but also by causing a very significant shrinkage of already developed LUAD, reducing tumor burden, cell proliferative rates and recruitment of CAFs, TAMs and T-lymphocytes in already preexisting tumors developed in adult mice expressing native levels of SOS1. Overall, these observations strongly suggest that the clinical benefit of targeting SOS1 is not limited to only blocking direct tumor growth, but also to reducing the various clinically deleterious effects of different stromal subpopulations of the TME in LUAD. It remains to be determined whether, and how much of, the therapeutic effect of SOS1 ablation is due to loss of activation of the resident KRAS$^{G12D}$ mutant proteins or the non-mutated KRAS$^{WT}$ proteins of the tumor/stromal cells. Although our assays could not discriminate the levels of specific WT

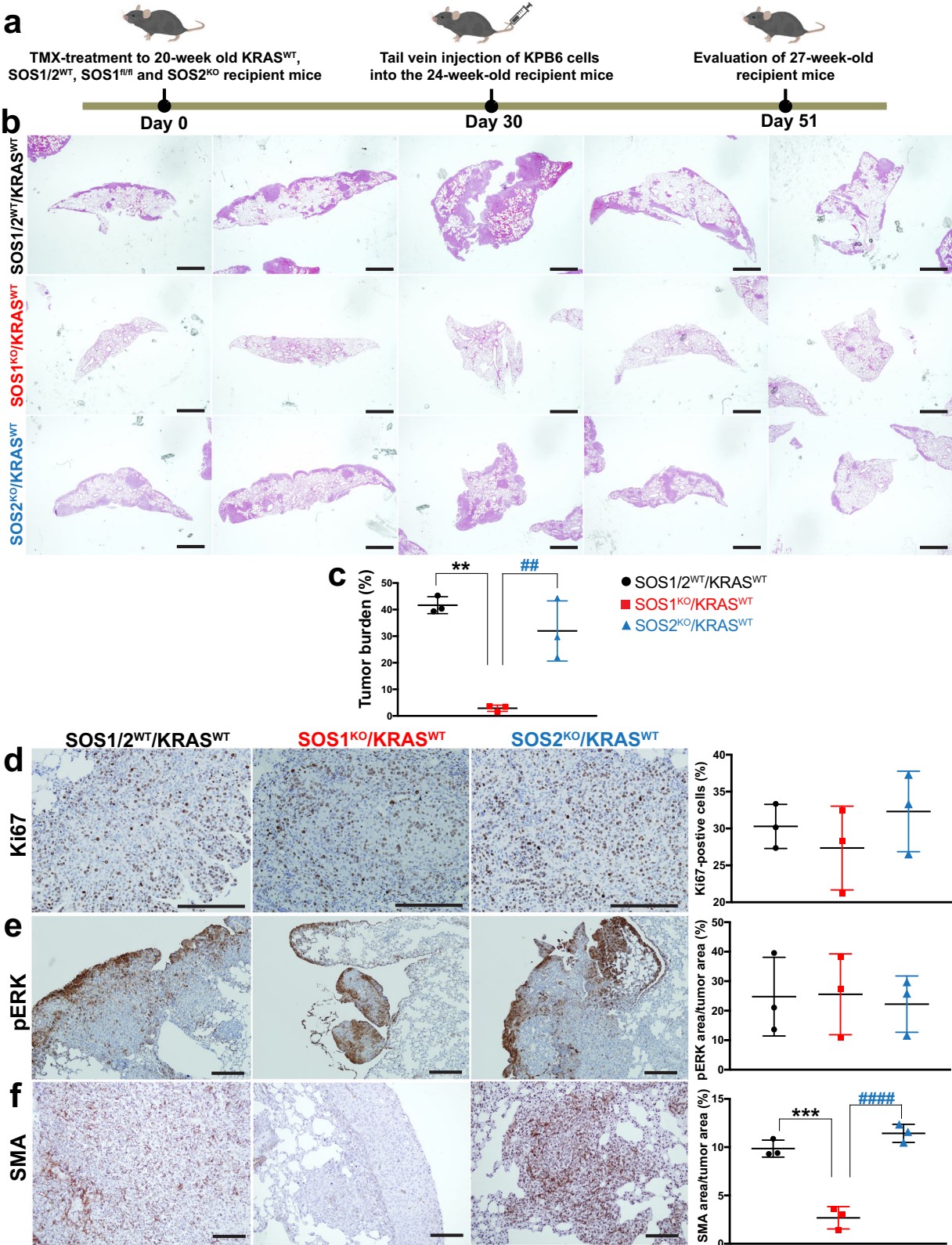

**Fig. 5 | Nesting and progression of KPB6 lung tumor cells in the lungs of syngeneic mice devoid of SOS1 or SOS2. a** Schematic illustration of the experimental regime and timing of tail vein injection of native KPB6 cells into mice of the indicated genotypes. **b** Representative images of H&E-stained, paraffin-embedded sections from the 5 different lobes of the lungs of TMX-treated (starting 1 month earlier), 27-week-old, KRAS[WT], SOS1/2[WT], SOS1[KO], and SOS2[KO]. **c** Bar graph quantifying the total lung tumor burden measured in each of the three experimental groups analyzed. **d–f** Representative images of paraffin-embedded sections from the lungs of 27-week-old, KRAS[WT], SOS1/2[WT], SOS1[KO], and SOS2[KO] after immunolabeling for Ki67 (**d**), pERK (**e**), and SMA (**f**). Scale bars: **b**, 1 mm; **d–f**, 100 μm. The bar graphs on the right side depict the percentage of Ki67-positive cells (**d**) in the lung tumor, as well as the percentage of total tumor area that was positively immunostained for pERK (**e**) or SMA (**f**). $n = 3$ independent experiments per genotype. Data shown as mean ± SD. * vs SOS1/2[WT] and # vs SOS2[KO]. In (**c**): **$P = 0.0011$ and ##$P = 0.0048$. In (**f**): ***$P = 0.0003$ and ####$P < 0.0001$. One-way ANOVA and Tukey's test Source data are provided as a Source data file.

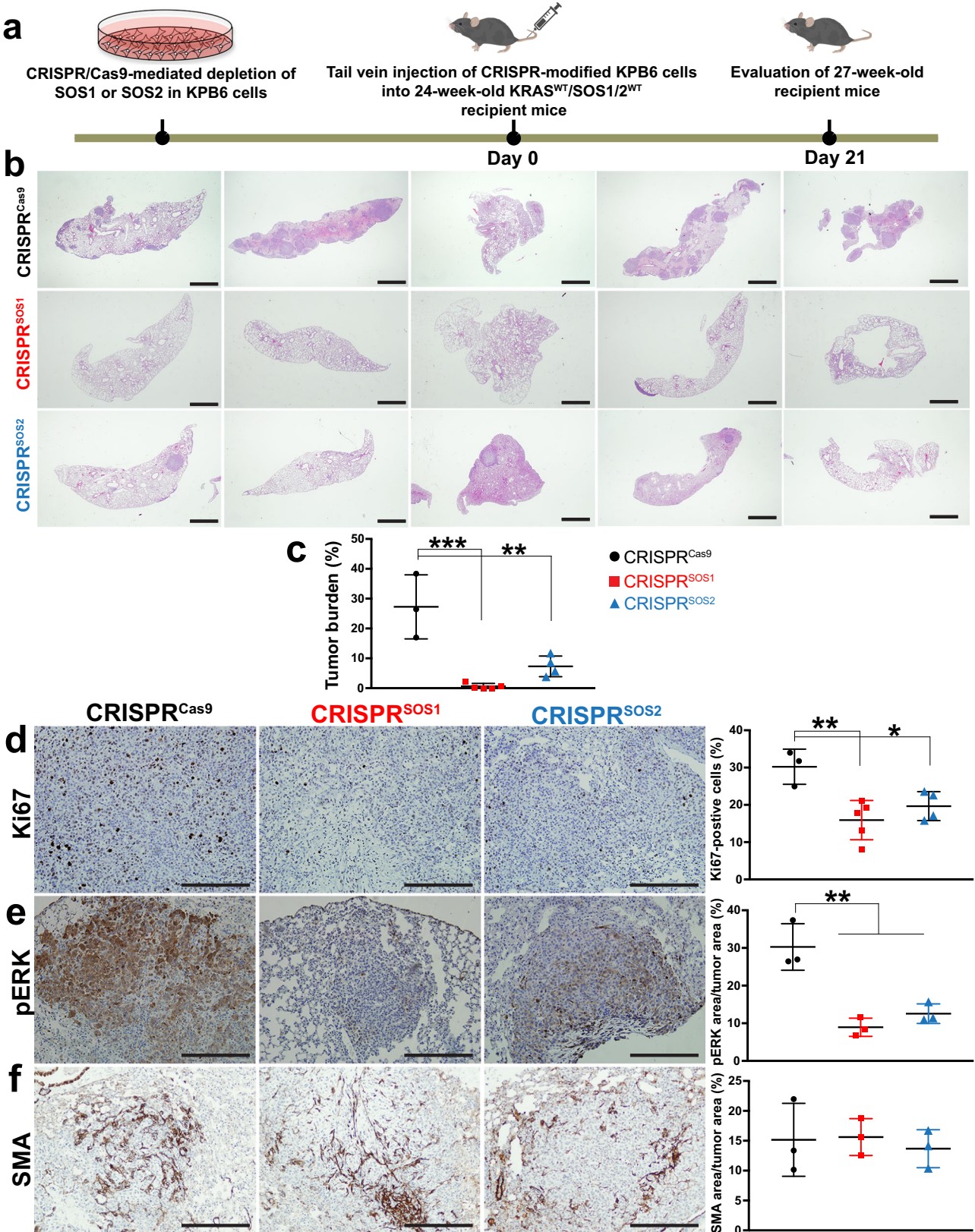

RAS activation in our mouse tumor models, a number of reports have consistently described a variety of critical functional contributions of the non-mutant WT RAS proteins to proliferation and/or transformation in RAS-mutated cancers that may range, depending on the specific tumorigenic context, from suppressing to promoting initiation or progression of tumor growth or even to fostering appearance of resistance to cancer drugs[12,13,16,17,53,54]. Comparing the effects produced by drugs acting preferentially on mutated[55] or on WT *KRAS* alleles[47,56] may help clarifying these issues in future.

We were also able to confirm the critical requirement of SOS1 and test the intrinsic or extrinsic nature of some inhibitory effects of SOS1 ablation on LUAD development by using an alternative experimental

**Fig. 6 | Nesting and progression of KPB6 lung tumor cells devoid of SOS1 or SOS2 in the lungs of WT syngeneic mice. a** Schematic illustration of experimental regime and timing of tail vein injection of CRISPR/Cas9-modified KPB6 cells devoid of SOS1 or SOS2 into wild-type KRAS$^{WT}$/SOS1/2$^{WT}$ mice. **b** Representative images of H&E-stained, paraffin-embedded sections from the 5 different lobes of lungs of 27-week-old, SOS1/2$^{WT}$/KRAS$^{WT}$ mice at 3 weeks after receiving intravenous injection of $1 \times 10^5$ CRISPR$^{Cas9}$, CRISPR$^{SOS1}$ and CRISPR$^{SOS2}$ KPB6 cells. **c** Bar graph showing the relative percentage of lung tumor burden in each experimental group. **d** Representative images of paraffin-embedded sections from lung tumors of 27-week-old wild-type mice SOS1/2$^{WT}$/KRAS$^{WT}$ that were injected with CRISPR$^{Cas9}$, CRISPR$^{SOS1}$ and CRISPR$^{SOS2}$ cells, after immunolabeling for Ki67. **d–f** The graphs on the right side show the relative percentage Ki67-positive cells (**d**), pERK levels (**e**), and SMA-immunopositive CAFs (**f**) in the tumors detected in each of the indicates genotypes and experimental conditions in the three experimental groups. Data shown as mean ± SD. In (**c**): ***$P$ = 0.0002 and **$P$ = 0.0026. In (**d**): **$P$ = 0.0062 and *$P$ = 0.040. In (**e**): **$P$ = 0.0017 vs SOS1$^{KO}$ and **$P$ = 0.0045 vs SOS2$^{KO}$. One-way ANOVA and Tukey's test. **c, d** $n$ = 3 independent mice for CRISPR$^{Cas9}$, $n$ = 5 independent mice for CRISPR$^{SOS1}$ groups and $n$ = 4 independent mice for CRISPR$^{SOS2}$ group. **e, f** $n$ = 3 independent mice for each genotype. Scale bars: **b** 1 mm and **d–f** 100 μm. Source data are provided as a Source data file.

approach in mice. Thus, by means of tail vein injection of a KRAS$^{G12D}$ LUAD mouse tumor cell line[43] into SOS1/2$^{WT}$, SOS1$^{KO}$, or SOS2$^{KO}$ mice we confirmed the specific reduction of lung tumor burden in SOS1$^{KO}$ recipient mice in comparison to equally injected WT or SOS2$^{KO}$ mice. In a converse experimental approach, the specific elimination of SOS1 from the KPB6 tumor cells[46] resulted also in significant reduction of lung tumor burden caused by injected, SOS1-silenced KPB6 cells in comparison to unmodified KPB6 cells. Of note, our observation of significant CAFs impairment in the TME when SOS1$^{KO}$ mice were injected with KPB6 cells, but not when WT mice were injected with SOS1/2$^{WT}$, SOS1$^{KO}$, or SOS2$^{KO}$ KPB6 cells, strongly suggests that the role of SOS1 in altering the TME in the recipient mice is separate from its impact on the growth of tumor initiating cells. Interestingly, silencing of SOS2 in the same experimental setting also led to reduce lung tumor burden, suggesting that SOS2 may also be able to contribute to intrinsic tumor growth initiation under these experimental conditions. In any case, the data in these experiments further confirms the critical relevance of SOS1 regarding the adequate homing and nesting of LUAD tumoral cells to their target lung organ.

Our experimental evidence supporting the critical contribution of SOS1 to LUAD development in mice is also consistent with the data on human LUAD available in various databases containing records from human lung tumor patients[49]. In particular, our computational analysis of the CRISPR library in the DepMap portal[48] uncovered a higher dependency rate on *hSOS1*, but not on *hSOS2*, for the proliferative ability of human LUAD cell lines, as well as a significant statistical correlation between *KRAS* expression and *hSOS1* (but not *hSOS2*) expression in those human tumor cell lines. Of note, the SOS1 DepMap dependency score calculated here for human LUAD cell lines appears to be clearly lower than the one previously reported for human CML cell lines, another tumor type where SOS1 is also critically required for malignant development[34]. Furthermore, using Cancertool[49] we also observed that *hSOS1* expression is significantly upregulated (as opposed to *hSOS2*) in human LUAD tumor samples in comparison to paired, non-tumor lung samples. In addition, LUAD patients with higher levels of *hSOS1* expression have lower survival rates than those with more reduced expression levels, whereas *hSOS2* expression level does not show any degree of statistical correlation with patient survival. Consistent with our current observations in lung cancer, other reports have recently described significant functional contributions of SOS1 in human hepatocellular carcinoma[57] and colorectal cancer[58].

In summary, the data presented in this report strongly indicate that SOS1 functionality is critically required for both the development of the LUAD tumor masses and for the generation of TME alterations that may also play significant pro-tumorigenic roles in LUAD development. Consequently, SOS1 constitutes a very relevant, actionable therapeutic target for LUAD whose inhibition is bound to produce clinical benefit not only at the level of the lung tumoral cells themselves but also at the level of different cellular subtypes in the TME.

Currently, the inhibition of SOS1 is already feasible by means of genetic silencing approaches like those used here and in other reports[9,59], as well as by pharmacological approaches using small-molecule drugs capable of modulating the GEF activity of SOS1 that were recently identified[21,60–64]. Interestingly, some of these inhibitors are thought to preferentially block non-mutated KRAS$^{WT}$[47,56,65] whereas others appear to predominantly target the action of SOS1 on specific KRAS mutant proteins[55]. In this regard, it is clearly apparent that the development of new drugs capable of modulating SOS1 and/or SOS2 activity in vivo is a highly desirable aim for the near future[4,10]. In any case, it will be highly interesting to use the same in vivo mouse models employed here to compare in future our current genetic ablation data with the pharmacological ablation of SOS1 mediated by recently developed inhibitors against this particular RAS-GEF[60–64]. Indeed, future studies of acute SOS1 pharmacologic inhibition in the mouse may help characterizing potentially evolving changes occurring in the lung cancer TME over time.

Finally, given the observed effects of SOS1 ablation on the TME, we might even postulate that SOS1 inhibition could be clinically beneficial not only for KRAS$^{G12D}$ LUAD but also for tumors driven by other RAS mutations or even by non-RAS-mutated tumors dependent from other drivers, such as mutant EGFR genes[10,36,64] capable of hyperactivating Ras ERK signaling in the lung. These considerations point also to the potential interest of therapeutic combinations adding SOS1 inhibition to a variety of already available inhibitory drugs (mostly kinase inhibitors)[10,21,56,62,63,66] that are currently used in the clinic against human LUAD.

## Methods

### LUAD animal model and analysis

The KRAS$^{LA2}$ mouse strain spontaneously developing KRAS$^{G12D}$-driven LUAD[43] was cross-mated to our tamoxifen (TMX)-inducible SOS1/2$^{KO}$ mouse system[26] to generate KRAS$^{G12D}$-expressing mice of the relevant SOS genotypes (SOS1/2$^{WT}$, SOS1$^{KO}$, SOS2$^{KO}$). The maximum size for an individual tumor was established in 1.5 mm in diameter. In addition, the estimation of lung tumor burden was monitored by signs of respiratory distress. All mice (no gender selected) were kept on the same C57BL/6J background and maintained under identical experimental conditions. The proper genotypes were monitored by PCR as described[26,43]. When indicated, TMX -containing chow diet[26] was administered to all experimental groups to avoid off-target effects. The animals were housed in cages with adequate space, bedding material for comfort and maintained under specific pathogen-free conditions, while maintaining 12-h dark/light. The ambient temperature was kept within 20–24 °C, and humidity levels ranged from 45–65%. Mice were kept, managed, and sacrificed in the NUCLEUS animal facility of the University of Salamanca according to European (2007/526/CE) and Spanish (RD1201/2005 and RD53/2013) legislations. All experiments were approved by the Bioethics Committee of the Cancer Research Center (#596).

To analyze the role of SOS1/2 proteins in KRAS$^{G12D}$-driven lung tumor initiation and progression, 1-month-old animals SOS1/2$^{WT}$, SOS1$^{fl/fl}$ and SOS2$^{KO}$ mice harboring KRAS$^{G12D}$ mutation were fed with TMX and then sacrificed at the indicated timepoints within 1–12 months of age. The number of visible surface tumors was quantified in animals euthanized at 1, 3, and 5 months of age. For analysis, lungs were dissected, fixed in 4% paraformaldehyde (PFA) for 24 h, and paraffin-embedded. To evaluate the effect of SOS1/2 depletion on preexisting tumors, 4-month-old, TMX-untreated SOS1/2$^{WT}$/KRAS$^{G12D}$ and SOS1$^{fl/fl}$/KRAS$^{G12D}$ mice were TMX-fed for 2 months before analysis.

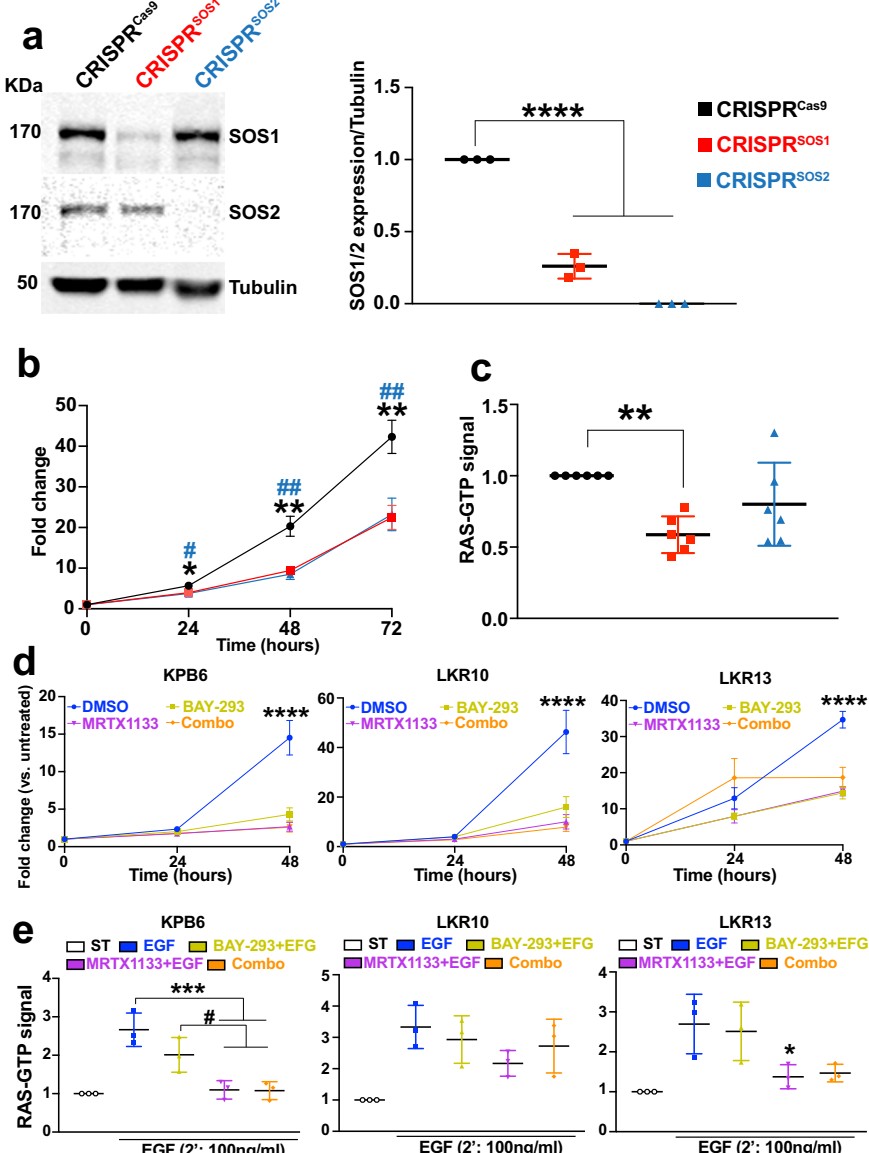

**Fig. 7 | Effects of CRISPR-mediated SOS1 or SOS2 depletion in KPB6 cells.**
**a** Representative WB images of lysates from KPB6 control cells expressing Cas9 (CRISPR[Cas9]) and from derived, SOS1-silenced (CRISPR[SOS1]) and SOS2-silenced (CRISPR[SOS2]) cells, immunolabelled for SOS1, SOS2 or tubulin. The bar graph quantitates the relative expression level of SOS1 or SOS2 normalized to that of tubulin in each case. $n = 3$ independent samples per group. Data shown as mean ± SD. ****$P < 0.0001$ vs CRISPR[SOS1/2]-depleted KPB6 cells. One-way ANOVA and Tukey's test. **b** Graph displaying growth curve of KPB6 cell cultures at 24, 48, and 72 h upon CRISPR-induced SOS1 or SOS2 ablation. $n = 12$ independent samples for CRISPR[Cas9] and $n = 8$ independent samples for CRISPR[SOS1/2] groups. Data shown as mean ± SD. One-way ANOVA and Tukey's test. At 24 h (*$P = 0.00379$ vs CRISPR[SOS1] and #$P = 0.00241$ vs CRISPR[SOS2]); at 48 h (**$P = 0.0020$ vs CRISPR[SOS1] and ##$P = 0.0013$ vs CRISPR[SOS2]) and at 72 h (**$P = 0.0022$ vs CRISPR[SOS1] and ##$P = 0.0078$ vs CRISPR[SOS2]). **c** Relative levels of RAS-GTP in cellular extracts from steady-state cultures of KPB6 cells upon CRISPR-mediated SOS1 or SOS2 depletion. $n = 6$

independent samples per group. Data shown as mean ± SD. **$P = 0.0038$ vs CRISPR[SOS1]. One-way ANOVA and Tukey's test. **d** Proliferative rates of cultures of the indicated KRAS[G12D]-mutated cell lines upon treatment with DMSO (vehicle), BAY-293, MRTX1133 or combo (BAY-293 + MRTX1133) for the times indicated. $n = 5$ independent samples per group. Data shown as mean ± SD. ****$P < 0.0001$ vs drug-treated cells. One-way ANOVA and Tukey's test. **e** Relative levels of RAS-GTP in extracts from serum-starved cultures of the indicated cell lines (white boxes) as well as EGF-stimulated cultures of the same cell lines after undergoing pretreatment with solvent vehicle (blue) or the indicated drugs, singly or in combination. $n = 3$ independent samples per group. Data shown as mean ± SD. *$P < 0.05$ ($P = 0.0498$ in LKR13 cells) or ***$P < 0.001$ (in KPB6 cells, $P = 0.0009$ and $P = 0.0008$, for MRTX + EGF or combo-treated groups, respectively) vs vehicle-treated, EGF-stimulated group, and #$P < 0.05$ ($P = 0.0354$ and $P = 0.0316$, for MRTX + EGF or combo-treated groups, respectively) vs BAY293 + EGF group. One-way ANOVA and Tukey's test. Source data are provided as a Source data file.

For in vivo microCT scanning analysis of tumor initiation and progression, 1-month-old mice of relevant genotypes (SOS1/2[WT]/ KRAS[G12D]; SOS1[fl/fl]/KRAS[G12D], and SOS2[KO]/KRAS[G12D]) were anesthetized and imaged using the SuperArgus microCT (Sedecal, Spain). Images were taken with 720 plane projections, 100 ms exposure time per projection, and X-ray energies of 45 kVp and 400 µA. Images were reconstructed and converted to 3D volumes using microCT Sedecal

ACQ software and tumoral and non-tumoral segmentations were performed by using the 3D Slicer image computing platform. TMX was then applied with feeding and the same procedure was performed in the very same animals at 3 and 5 months of age. For studies of tumor regression, 4-month-old, TMX-untreated, SOS1[fl/fl]/KRAS[G12D] mice were fed with TMX for two additional months (to induce SOS1 ablation), and in vivo images of the very same animals (at 4 months

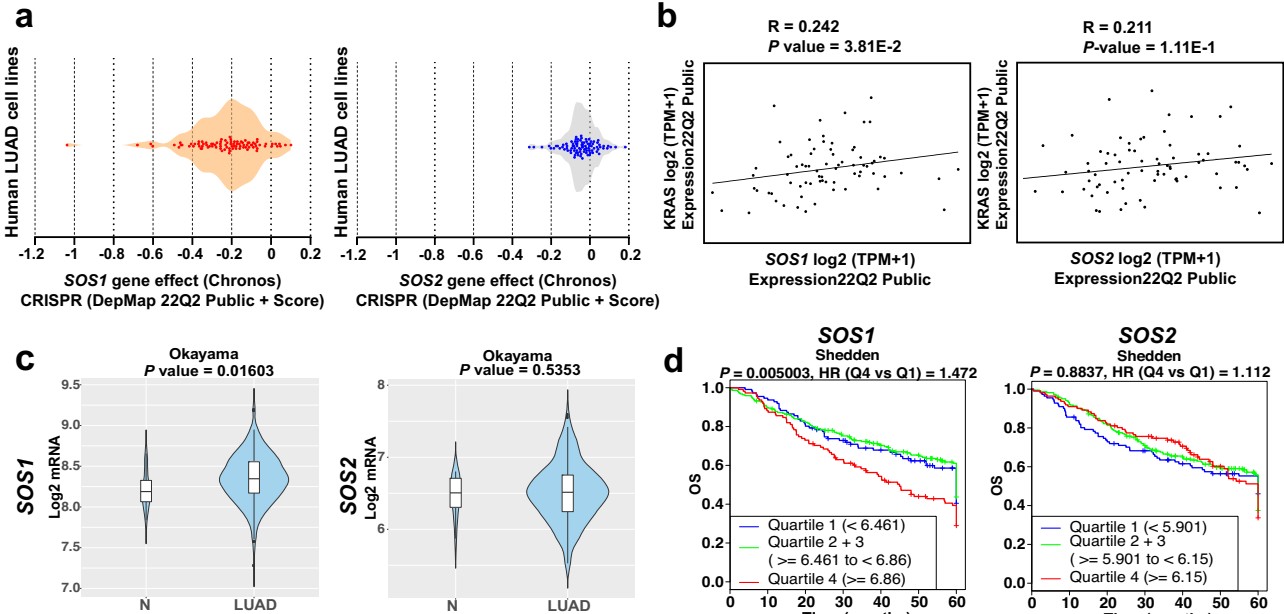

**Fig. 8 | In silico assessment of the functional relevance of SOS1 for human LUAD. a** *hSOS1* and *hSOS2* gene dependency score of human LUAD cell lines featured in the CRISPR library of the DepMap portal. **b** Evaluation of the correlation between *hKRAS* gene expression level and that of *hSOS1* and *hSOS2* in human LUAD cell lines gathered in the DepMap portal library. Pearson's correlation (R) and *P* values are shown in each graph. **c** Violin plots comparing *hSOS1* and *hSOS2* gene between non-tumoral (N; cohort size: 20) and LUAD specimens (cohort size: 226) in the Cancertool Okayama dataset. The Y-axis represents the Log2-normalized gene expression (fluorescence intensity values for microarray data or, sequencing reads values obtained after gene quantification with RSEM and normalization using Upper Quartile in case of RNAseq). The white boxes represent the interquartile range of the data. Minima and maxima in the boxes indicate the first quartile and

the third quartile, respectively, whereas the center indicates the median. The whiskers represent the upper and the lower adjacent values. Outside points (traditionally classified as mild and severe outliers) can be also observed. A Student t-test is performed in order to compare the mean gene expression between two groups. **d** Kaplan–Meier curves showing the overall survival (OS) of patient groups selected in the Cancertool Shedden dataset, according to the quartile expression of the *hSOS1* or *hSOS2* genes. Quartiles represent ranges of expression that divide the set of values into quarters. Quartile color code: Q1 (Blue), Q2 + Q3 (Green), Q4 (Red). Each curve represents the percentage (Y-axis) of the population that exhibits recurrence of the disease along time (X-axis, in months) for SOS1 or SOS2 expression distribution quartile. Vertical ticks indicate censored patients. Quartile color code: Q1 (blue), Q2 plus Q3 (green), Q4 (red).

and 6 months of age) were compared. Images were taken at the NUCLEUS Molecular Imaging Laboratory of the University of Salamanca.

Physiological parameters ($O_2$ saturation, respiratory rate, EKG) were monitored in 12-month-old, TMX-treated, SOS1/2^WT/KRAS^G12D; SOS1^KO/KRAS^G12D and SOS2^KO/KRAS^G12D mice using a Small Animal Physiological Monitoring System (Cat# 75-1500, Harvard Apparatus) following manufacturer's instructions. As controls of normal physiological values, 12-month-old, TMX-treated, SOS1/2^WT/KRAS^WT mice were employed.

The expression level of 40 different cytokines was measured in lung tumors from 5-month-old, TMX-treated, SOS1/2^WT/KRAS^G12D, SOS1^KO/KRAS^G12D, and SOS2^KO/KRAS^G12D mice using the Mouse Cytokine Array Panel A (Cat#ARY006, R&D Systems) as per manufacturer´s instructions.

**Histology and immunostaining**
Paraffin-embedded lung samples were cut and stained with hematoxylin and eosin or Masson's trichrome according to standard procedures. Pathological grade classification of tumor samples comprised hyperproliferating type II pneumocytes, adenomas and carcinomas. For immunohistochemistry, sections were dewaxed, microwaved in citrate buffer (pH 6) and incubated overnight with anti-pERK1/2 (Cat#9101, 1:500, Cell Signaling), anti-Ki67 (Cat# MAD-000310QD-3, 1:50, Master Diagnostica), anti-SMA (Cat# A5228, 1:500, Sigma), anti-CD68 (Cat# ab125212, 1:100, Abcam) and anti-CD3 (Cat# ab5690, 1:50, Abcam) at 4 °C. Sections were then incubated with biotin-conjugated secondary antibodies followed by Vectastain Elite ABC reagent and the reaction product visualized by incubating the sections in 0.025% 3.3'-

diaminobenzidine and 0.003% $H_2O_2$ in PBS. The processing and staining of the sections were performed by the PMC-BEOCyL Unit (Comparative Molecular Pathology-Biobank Network of Oncological Diseases of Castilla y León).

**RT-qPCR assays**
Proper *SOS1* (upon TMX-administration) or *SOS2* gene silencing was evaluated by qPCR. Lungs from TMX-treated mice from of the three experimental groups were isolated and homogenized with NZYol (NZYTech Cat# MB18501, Lisbon, Portugal). RNA was isolated from the NZYol lysates following manufacture's indications, and further purified using RNeasy Mini Kit columns (Qiagen, Cat# 74004). RNA concentration and quality was then assessed by RNA capillary electrophoresis columns (Agilent Technologies, Santa Clara, USA). RT-qPCR assays to detect expression levels of *SOS1* (Fw 5'-GAGCCAAACACGA-GAGACAC-3'; Rv 5'-ATTCTGCACTGCTAGCACCA-3') and *SOS2* (Fw 5'-AGTGAGTGCAGTCAACTCCG-3'; Rv 5'-GTGGTCCTGACTTAGTTCCA-3' were performed using the Luna Universal One-Step RT-qPCR kit (New England Biolabs, Cat# E3005L. Ipswich, MA, USA) following the manufacture's protocol, and *β2-MICROGLOBULIN* (Fw 5'-ATGGG AAGCCGAACATACTG-3'; Rv 5'-CAGTCTCAGTGGGGGTGAAT-3') was used as a housekeeping gene to normalize results.

**LUAD cell lines**
LKR10 and LKR13 cells are mouse lung cancer cell lines holding KRAS^G12D mutation and they were derived by serial passage of minced LUAD tissues from two tumors isolated from separate lobes of the same Kras^LA1 mouse[43]. KPB6 cells are LUAD cells harboring KRAS^G12D and p53 mutations[46].

## CRISPR/Cas9-mediated depletion of SOS1 and SOS2 in KPB6 tumor cells

KPB6[46] and HEK293T (Cat#CRL-3216, ATCC) cells were grown in Dulbecco's modified Eagle's medium (DMEM) (4.5 g/L glucose +10% FBS + 1% penicillin/streptomycin). Plasmids pLV[Exp]-Puro-TRE>hCas9 (VB010000-0366ycc), pLV[2CRISPR]-Hygro-U6>mSos1 (VB210819-1116tch) and pLV[2CRISPR]-Hygro-U6>mSos2 (VB210819-1114mxf) were purchased from VectorBuilder Inc (Neu-Isenburg, Germany). To generate lentiviral particles, plasmids were individually co-transfected with packaging vectors (Cat#RV111, pCMV-Gag-Pol and Cat#RV110, pCMV-VSV-G, CellBioLabs) into HEK293T cells as previously described[67]. Briefly, HEK293T cells were used to produce the lentiviral vector containing Human codon-optimized CRISPR-associated protein 9 from Streptococcus pyogenes (VB010000-0366ycc). $3 \times 10^6$ HEK293T cells were seeded the day before transfection. HEK293T cells were changed to free-antibiotics DMEM and the lentiviral vector assembly formulation (1 mL Opti-MEM medium, Cat# A4124802, Gibco), 12.5 μg of CRISPR-hCas9 plasmid, 9.5 μg pCMV-GAG-POL, 3 μg pCMV-VSV-G and 62.5 μL of JetPEI® transfection reagent (Cat# 101000053, Polyplus) was added per each p100 plate. This mix was incubated for 20 min at room temperature (RT) before gently adding it to the culture plate. After 48 h, supernatant was collected, centrifuged (5 min, 1500 rpm), and filtered (0.45 μm pore size) to obtain filtered virus. KPB6 cells were trypsinized and seeded at a density of $1 \times 10^5$ cells. KPB6 cells were infected with 2 mL of filtered virus with Polybrene® (8 μg/mL, Cat#sc-134220, Santa Cruz Biotechnology) and centrifuged at 800 g for 30 min at RT. 24 h after reinfection, Puromycin (2 μg/mL, Sigma) was supplemented to the culture media for positive selection. Stable KPB6-Cas9 cells were reinfected with pLV[2CRISPR]-U6 with two gRNA for deletion of SOS1 (VB210819-1116tch) and SOS2 (VB210819-1114mxf), and selected with Hygromycin (100 μg/mL).

To assess cell proliferation, AlamarBlue™ Cell Viability Reagent (Cat.DAL1100, Invitrogen) was added (10 μL/well) and incubated in dark for 1 h at 37 °C and 5% $CO_2$. Fluorescence intensity was then evaluated at 590 nm using TECAN Infinite® 200 PRO microplate reader taking measurements at 0, 24, 48, and 72 h post-doxycycline induction. To determine the effect of the SOS1::KRAS interaction inhibitor BAY-293[47] or the $KRAS^{G12D}$ inhibitor MRTX1133[68] (individually or combined), $1 \times 10^6$ KPB6$^{Cas9}$ cells were treated with BAY-293 (1 μmol/L), and/or MRTX1133 (5 nmol/L), and cell proliferation quantitated as described above.

Intracellular RAS-GTP levels in steady-state, CRISPR-modified KPB6 cell cultures (CRISPR$^{Cas9}$, CRISPR$^{SOS1}$, CRISPR$^{SOS2}$) were measured using the RAS G-LISA™ assay (#BK131, Cytoskeleton, Inc) as recommended by the manufacturer. The effect of BAY-293 and MRTX133 on RAS activation was also evaluated in serum-starved KPB6$^{Cas9}$, LKR10 and LKR13 cells treated with vehicle (DMSO), BAY-293 (4 μmol/L), and/or MRTX1133 (5 nmol/L) for 2 h and stimulated with EGF (100 ng/mL) for 2 min.

The impact of SOS1/2 depletion on LUAD development was also evaluated in an orthotopic mouse tumor model involving intravenous tail injections of native or CRISPR/Cas9-modified KPB6 LUAD cells. To investigate the influence of SOS1/2 ablation only in the lung stroma, $1 \times 10^5$ native KPB6 cells were injected in the tail vein of 24-week-old, TMX-treated, SOS1/2$^{WT}$/KRAS$^{WT}$, SOS1$^{KO}$/KRAS$^{WT}$, and SOS2$^{KO}$/KRAS$^{WT}$ mice. To test the influence of SOS1/2 ablation only in the LUAD tumor cells, $1 \times 10^5$ CRISPR/Cas9-modified KPB6 cells (CRISPR$^{Cas9}$, CRISPR$^{SOS1}$, and CRISPR$^{SOS2}$) were injected in the tail vein of wild type, 24-week-old (SOS1/2$^{WT}$/KRAS$^{WT}$) animals. In all cases, mice were sacrificed 3 weeks after the injection and their lungs dissected and fixed in PFA for 24 h for later analysis.

## In silico human LUAD analyses

DepMap identifies cancer vulnerabilities by identifying genetic dependencies in different types of human tumors[48]. The specific gene dependency scores of *hSOS1* and *hSOS2* in human LUAD cell lines were calculated using the DepMap 22Q2 Public + Score dataset in their web portal (https://depmap.org/portal/, accessed on 5 February 2023). A low dependency score indicates a higher likelihood that the gene of interest is essential for a given tumor cell line.

The CANCERTOOL database and web-based computational tools (http://genomics.cicbiogune.es/CANCERTOOL/index.html, accessed 5 February 2023) was used for a comprehensive evaluation of the relevance of *hSOS1* and *hSOS2* gene expression data for LUAD development and survival[49].

## Western immunoblotting and pull-down assays

30 μg protein extracts from lung tumors isolated in the three experimental groups or CRISPR-modified KPB6 cells were electrophoresed and immunoblotted using the following primary antibodies and dilutions: anti-SOS1 (Cat#610096, 1:500, BD), anti-SOS2 (Cat#sc-15358, 1:500, Santa Cruz Biotechnology), anti-tubulin (Cat#T5293, 1:10,000, Sigma) and anti-vinculin (Cat#26520-1-AP, 1:5000, ProteinTech) and the corresponding secondary antibodies: Goat anti-mouse DyLight™ 800 (Cat#SA5-35521, 1:10,000, ThermoFisher Scientific) and Goat anti-rabbit Fluor® 680 (Cat#A21076, 1:5000, Invitrogen). To determine the levels of the GTP-bound levels of RAS in lung tumors (anti-RAS, Cat# 05-516, 1:1000, Millipore), we performed pull-down assays for active RAS as previously reported[27].

## Statistical analysis

GraphPad Prism 8.0.1 (GraphPad Inc., USA) software was used. Statistical significance was determined by one-way ANOVA using the Tukey's method to correct for multiple comparisons. For comparisons established only between two groups we used Student's two-tailed, unpaired t-test. To compare tumor regression in the very same animals, paired t-test was performed. Survival analysis was performed by the Kaplan–Meier method and between-group differences in survival were tested using the Log-rank (Mantel-Cox) test. *n* values mentioned in the figure legends indicate the number of independent animals/ samples used per experimental group. However, notice that our quantitative analyses always involved measurements of the total number of tumors present in all 5 lung lobes of each animal analyzed. Results are expressed as mean ± Standard Deviation (SD). Significant differences are considered at $P$ value < 0.05.

## Reporting summary

Further information on research design is available in the Nature Portfolio Reporting Summary linked to this article.

## Data availability

The authors declare that the data supporting the findings of this study are available within the paper and its supplementary information files. CANCERTOOL database (http://genomics.cicbiogune.es/ CANCERTOOL/) and DepMap portal library (https://depmap.org/ portal/) were used in this study. Some pictures were created with BioRender (https://www.biorender.com/). Source data are provided with this paper.

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

## Acknowledgements

Work supported by grants ISCIII-MCUI (FIS PI19/00934), JCyL (SA264P18-UIC 076), Areces Foundation (CIVP19A5942), and ISCIII-CIBERONC (group CB16/12/00352) to E.S.; Solorzano-Barruso Foundation (FS/32-2020) and Eugenio Rodriguez Pascual Foundation to F.C.B.; MCI (RTI2018-099161-A-I00) to E.C. This research was co-financed by FEDER funds. These CIC groups are supported by the Programa de Apoyo a Planes Estratégicos de Investigación de Estructuras de Investigación de Excelencia of Castilla y León autonomous government (CLC-2017-01) and AECC Excellence program Stop Ras Cancers (EPAEC222641CICS).

## Author contributions

Conception and design: F.C.B., E.C. and E.S. Development of methodology: F.C.B., R.G.-N., C.C., J.B., P.R.-R., R.F.-M., A.O.-S.J. and N.V. Acquisition of data (provided animals, acquired and managed patients, provided facilities, etc.): F.C.B., R.G.-N., N.C., J.B., E.C. and E.S. Analysis and interpretation of data (e.g., statistical analysis, biostatistics, computational analysis): F.C.B., R.G.-N., J.B., R.F.-M., E.C. and E.S. Writing, review, and/or revision of the manuscript: F.C.B., R.G.-N., J.B., R.F.-M., A.O.-S.J., N.V., E.C. and E.S.

## Competing interests

The authors declare no competing interests.
