## [Peer Review File · Nature Communications]

REVIEWER COMMENTS

Reviewer #1 (Remarks to the Author):

The authors crossed their previously generated Cre-inducible *Sos1* or *Sos2* null alleles into the *Kras^{LA2}* background and demonstrate, that loss of *Sos1*, and to a lesser extent the loss of *Sos2*, reduced tumor burden, extended lifespan, and when conditionally inactivated in established tumors, led to tumor regression. This is a really beautiful set of experiments -well powered, lots of confirmatory experiments, very convincing results, rather difficult in nature, used important clinical metrics, etc. Another highlight of the paper was the demonstration that the loss of *Sos1*, and less so or not at all the loss of *Sos2*, in both the stroma and tumor inhibited tumorigenesis through a series of really nice xenograft studies, either injecting an isogenic *Kras/p53* lung cancer cell line into *Sos1* or *Sos2* null backgrounds, or CRISPR/Cas9-mediated inactivation *Sos1* or *Sos2* in the same cell line and injecting it into wild-type mice. The authors then hit human expression databases to make the argument that *SOS1*, but not *SOS2*, is common essential in human LUAD cell lines and upregulated, which was associated with worse outcomes. The strengths of this study are the rigor of the mouse studies and the new findings that *Sos* promotes tumorigenesis through expression in both the stroma and tumor. Weaknesses are the somewhat expected results based on testing these genotypes already in other RAS-driven mouse models, the somewhat descriptive nature of the study (although to the authors credit, efforts were made to characterize signaling and tumor markers), and minor concerns in over-interpreting results. There is high enthusiasm for this study, particularly because of the comprehensive nature of the genetic studies and the nice experiments parsing out stroma and tumor contributions for *Sos* proteins, with only minor concerns, none of which detract in a significant way for the overall conclusion of the work.

Minor concerns

1. The novelty of this work is diminished by a study by You et al., published in the journal *Blood* in 2018 demonstrating reduced early leukemogenesis and a longer lifespan in *Vav-Cre;Sos1^{fl/fl};Kras^{LSL} G12D/+* mice. Similarly, the authors themselves demonstrated that loss of *Sos1* compared to *Sos2* had a greater effect on reducing oncogenic *Hras*-driven solid tumorigenesis induced by DMBA/TPA carcinogenesis in their wonderful study published in 2018 in the journal *Molecular & Cellular Biology*. Perhaps the authors could make more in-depth comparisons and contrasts with of these studies in their discussion.

2. The sentence “The quest for new therapies must also take into account the experimental evidence demonstrating that the activity of non-mutated (WT) cellular RAS proteins is critically required for the development of oncogenic RAS-driven tumors...” does not entirely capture the complexity of wild-type RAS contributions to cancer. On one hand, Zang et al., demonstrated in a *Nature Genetics* paper published in 2001 that *Kras^{+/-}* mice are more susceptible to urethane carcinogenesis while Lim et al., demonstrated in a *Nature* paper published in 2008 that knockdown of wild-type RAS inhibited xenograft tumor growth of pancreatic cancer cell lines. One interpretation of these opposite results is that wild-type RAS proteins inhibit tumor initiation but promote later stages of tumorigenesis. Nevertheless, the role of wild-type RAS proteins in tumorigenesis is more complicated than described by the above the

authors. Given that the authors study an activator of wild-type RAS, could they please provide a more comprehensive review of the literature, which will better set the stage for why their current work is important.

3. The authors describe, rather than state their quantification. For example, “Quantitation of the number of tumors visible on the pleural surface of the lungs of 1-5 month-old mice during these early months of tumor initiation and development in this mouse model 38 showed that the total tumor count in SOS1KO/KRASG12D and SOS2KO/KRASG12D mice was dramatically reduced in comparison to their control SOS1/2WT/KRASG12D counterparts.” Rather than using descriptors, could the authors please first report the actual quantification, then layer in their descriptors.

4. Please elaborate the separation of the two roles of Sos1. Significant decrease in CAFs in tumor microenvironment was observed when Sos1-KO mice were injected with KPB6 cells (Figure 4) but not when WT mice were injected with Sos1/2-WT, Sos1-KO, or Sos2-KO KPB6 cells (Figure 5). This suggests that the role of Sos1 in altering the tumor microenvironment in the recipient mice is separate from its impact on the growth of tumor initiating cell.

5. Please clarify what tissues Cre excision of Sos1 and Sos2 occurs in, and how this ablation of the genes was confirmed.

6. The authors state that a “critical requirement of SOS1 for in vivo initiation, maintenance, and progression of KrasG12D-driven LUAD in mice” but this is supported by a single genetic background and a single tested LUAD murine cell line. Perhaps the authors could state what they demonstrate, then speculate on the wider possibilities of their findings in regard to lung cancer in general. This is somewhat of a reoccurring theme, for example, the experimental studies are all performed in mice, so perhaps noting this caveat when speculating on the role of SOS proteins in human cancer is warranted.

7. The authors introduce targeting SOS1 as a therapeutic strategy partially due to its important role in activation of the WT KRAS, but there was no further exploration of the impact of genetic ablation of Sos1 on the WT Kras activity or signaling. Please either measure WT Kras/Nras/Hras GTP loading in the tumor setting or indicate the caveat to this conclusion is that the level of wild-type RAS activation was not determined as SOS can activate other small GTPases.

8. The in silico data in the final figure does not suggest dependency of LUAD cell lines on SOS1, but rather a slightly higher dependency on SOS1 than SOS2. In DepMap, the dependency of LUAD cells on KRAS or SHP2 is much higher than SOS1, but another RAS-GEF RASGRP2 is akin to SOS1. Please consider modifying the text to clarify the in silico analysis.

9. The authors previously demonstrated that Sos1, but not Sos2, was critical for early thymocyte maturation in their Molecular & Cellular Biology paper from 2013. Here, they beautifully show a decrease in CAFs, TAMs, and T-lymphocytes in TME with Sos1, but not Sos2, ablation. Could the authors please provide data demonstrating that TME changes were not due to the hematopoietic composition changes in the whole animal and, rather, are specific to the TME?

10. Replicating the growth impairment phenotype with genetic ablation and pharmacological inhibition in Figure 7 with another KrasG12D-mutant LUAD cell line would help strengthen the claims regarding a KrasG12D-mutant LUAD-specific SOS1 dependency.

Reviewer #2 (Remarks to the Author):

In this work, Baltanás and colleagues describe the generation of KRAS mutant/SOS1-2 KO genetic engineered mouse model of lung cancer.

KRAS targeting has become particularly relevant since the approval of KRAS G12C covalent inhibitors in clinic and SOS1 inhibitors are in clinical testing meaning that the development of these in vivo models is very valuable.

I would point out to the authors that the concept of constitutively activation of mutant KRAS is bypassed by the more recent data showing that KRAS mutant alleles are subjected to GEF-mediated activation, GAP-mediated inactivation, and interaction with downstream effectors in different ways (for example PMID: 23487764 and PMID: 26037647 and other works on KRAS G12C inhibitors-based combinations), and the contribution of RAS wt isoforms is not the main or the unique source of RAS pathway re-activation upon RAS inhibition. Having said that, I appreciated the description provided in this manuscript both at the tumor intrinsic and tumor extrinsic levels.

The mainly cytostatic effect of SOS1 KO (and to a lesser extent, SOS2 KO) is interesting and suggests that SOS1 inhibition should be combined with other inhibitors to successfully induce cell death, as previously reported. This makes these GEMM models very useful on the long term to study acquired resistance.

I do not particularly like the human validation as final part of the work. In my opinion, these kinds of analyses should be presented at the very beginning to further highlight the importance of SOS1 role in the RAS-targeting and not to show that human data replicate what happen in mice. I would suggest moving these data at the beginning, merged with figure 1. Also figure 8D is missing the SOS1/SOS2 labeling.

Regarding the tumors that develop in the SOS1 KO model, did authors conducted further analyses? Is RAS pathway active? Is it mediated by SOS2? If these tumors can be grown in vitro, would SOS2 silencing cause cell death? Or direct RAS inhibition would work better?

Given the relevance of the journal and topic, I think some in vivo experiments using pharmacological treatments would be beneficial. I would be curious to know if concomitant G12D inhibition would synergize with SOS1 ablation and/or if SOS1 inhibition would synergize with SOS2 KO. As for now, I believe that this is the biggest lack in this story.

Reviewer #3 (Remarks to the Author):

The manuscript by Baltanas et al uses genetic approaches to assess the independent roles of SOS1 and SOS2 in KRASG12D-driven lung adenocarcinoma (using the Tyler Jacks LA2 strain that gives rise to spontaneous LUAD). The authors show that each RASGEF plays a role in tumorigenesis, however, the effect of Sos1 KO was much more prominent than for Sos2 KO. For SOS2, the authors show a role in tumor initiation: Young Sos2 KO mice (1-5 months) show a lower number of surface tumors and a lower overall tumor burden compared to WT controls, but these differences become less pronounced as mice age so that the overall effect on survival is not significant (although the differences in survival at < 200 days look convincing). This effect on tumor initiation is even more pronounced in adoptive transfer experiments where SOS2 was deleted using CRISPR (See Fig. 6), where the effects were more similar to SOS1 KO.

For SOS1, the overall effects on tumorigenesis were more pronounced. The authors use *Sos1^{f/f}* mice where the floxed allele is not removed until 1 month of age and show a more pronounced early and late reduction in tumor burden compared to *Sos2* KO, along with a significant enhancement in overall survival. *Sos1* KO in older mice (4 months) also caused partial regression of established tumors, showing that SOS1 is a potential therapeutic target. The authors further show that these effects effect of SOS1 ablation on tumorigenesis were due both reduced tumor cell proliferation (reduced Ki67 and pERK staining) and on the tumor microenvironment (reduced SMA staining for cancer-associated fibroblasts). The authors confirm these two roles for SOS1 in adoptive transfer experiments: transfer of *Sos*^{WT} tumor cells into *Sos1* KO recipient mice showed reduced tumor burden that was due to low numbers of CAFs (tumor extrinsic); transfer of *Sos1* KO cells into WT recipient mice showed reduced tumor burden (tumor intrinsic). In contrast, SOS2 only showed tumor intrinsic effects. Finally, the authors probed both DepMap and CANCEERTOOL databases to show a correlation between SOS1 expression and cell line (DepMap) or patient (CANCEERTOOL) survival.

This is a fantastic paper, and the authors should be commended for their tremendous work. The experiments are well thought out, and the conclusions are all supported by the data. While I have several specific suggestions that will help bolster the authors' arguments, I do not think any additional experimentation is needed (well done by all).

Comments

1. In the tumor regression experiments (Fig. 4), it is unclear as the data are currently presented whether the reduced tumor burden is due to true regression or only in stasis (inhibition of further proliferation). The authors present tumor volumes at 6 months...however they should (per the methods) have paired 4-month measurements for each mouse. For this data it is important for the authors to show these paired measurements -preferably for each mouse. This would be a paired statistic showing 4 and 6

month measurements in the *Sos1^{f/f}* mice and the *Sos1KO* mice. For an example in an unrelated field see Lake et al, *Cell Death and Disease* 12:400 (Fig. 1B-D). This is especially important given the lack of differences in their cell death (CC3) staining.

2. In the discussion (lines 376-379) the authors state ‘it remains to be determined whether ...the effect of *SOS1* ablation is due to loss of activation of the resident *KRASG12D* mutant proteins or the non-mutated *KRASWT* proteins. I submit that if the effects of *SOS1* ablation is on WT proteins, it is very likely that inhibition of WT *HRAS* and WT *NRAS* in both the tumor and the stroma are significantly important as well. There is substantial evidence for the role of WT *RAS* family members, distinct from the mutant allele, in mutant *KRAS*-driven tumorigenesis. This should be expanded on in the discussion. This comes up again on line 422 – where *SOS1* inhibitors also block WT *HRAS* and *NRAS*.

3. On lines 381-386, the adoptive transfer into *Sos1KO* recipients argues for a non-cell autonomous function...the authors should be careful to clarify the discussion here.

4. In discussing the adoptive transfer of *Sos1 KO* versus *Sos2 KO* cells (lines 390-392), the authors hypothesize that the reduced tumor burden in *Sos2 KO* was due to altered cell homing to the heart. Is there data supporting this? Alternatively, this could simply reflect the role of *SOS2* in tumor initiation but not in the TME.

5. The authors seem to miss an opportunity to really hammer the point that *SOS1* effects are in both the tumor and in CAFs. Indeed, the authors previously showed a prominent role for *SOS1* in proliferation of MEFs...these data support the CAF data.

6. In the first paragraph of the discussion, the authors have repeated “uncovered also a measureable delaying effect of *SOS2* ablation regarding the onset and initial stages of LUAD development in *KRASG12D* mice” on lines 325-327 and 328-330.

7. In Fig. 2B, it appears that *Sos2 KO* also showed reduced pERK relative to WT. Is this significant?

REVIEWER #1 (Remarks to the Author):

The authors crossed their previously generated Cre-inducible Sos1 or Sos2 null alleles into the KrasLA2 background and demonstrate, that loss of Sos1, and to a lesser extent the loss of Sos2, reduced tumor burden, extended lifespan, and when conditionally inactivated in established tumors, led to tumor regression. This is a really beautiful set of experiments -well powered, lots of confirmatory experiments, very convincing results, rather difficult in nature, used important clinical metrics, etc. Another highlight of the paper was the demonstration that the loss of Sos1, and less so or not at all the loss of Sos2, in both the stroma and tumor inhibited tumorigenesis though a series of really nice xenograft studies, either injecting an isogenic Kras/p53 lung cancer cell line into Sos1 or Sos2 null backgrounds, or CRISPR/Cas9-mediated inactivation Sos1 or Sos2 in the same cell line and injecting it into wild-type mice. The authors then hit human expression databases to make the argument that SOS1, but not SOS2, is common essential in human LUAD cell lines and upregulated, which was associated with worse outcomes. The strengths of this study are (1) the rigor of the mouse studies and (2) the new findings that Sos promotes tumorigenesis through expression in both the stroma and tumor. Weaknesses are (1) the somewhat expected results based on testing these genotypes already in other RAS-driven mouse models, (2) the somewhat descriptive nature of the study (although to the authors credit, efforts were made to characterize signaling and tumor markers), and (3) minor concerns in over-interpreting results. There is high enthusiasm for this study, particularly because of the comprehensive nature of the genetic studies and the nice experiments parsing out stroma and tumor contributions for Sos proteins, with only minor concerns, none of which detract in a significant way for the overall conclusion of the work.

Reply. We gratefully acknowledge this reviewer's high enthusiasm for our study, as well as her/his overall really positive assessment of the work and conclusions included in the manuscript. Our responses to the Minor concerns mentioned in this reviewer's report are provided below:

Minor concerns

1. *The novelty of this work is diminished by a study by You et al., published in the journal Blood in 2018 demonstrating reduced early leukemogenesis and a longer lifespan in Vav-Cre;Sos1^{fl/fl};Kras^{LSL} G12D/+ mice. Similarly, the authors themselves demonstrated that loss of Sos1 compared to Sos2 had a greater effect on reducing oncogenic Hras-driven solid tumorigenesis induced by DMBA/TPA carcinogenesis in their wonderful study published in 2018 in the journal Molecular & Cellular Biology. Perhaps the authors could make more in-depth comparisons and contrasts with of these studies in their discussion.*

Reply. We certainly agree with this reviewer's comment noting that our current observations on SOS and KRAS-LUAD could be somewhat expected based on prior testing of these genotypes in other RAS-driven mouse models such as, particularly, the study by You et al demonstrating the unique dependence on SOS1 in KRAS^{G12D}-induced leukemogenesis.

Following the reviewer's advice, we have now corrected and extended a sentence (lines 360-362) of the Discussion remarking the consistency of our current data about the effect of SOS1 ablation in KRAS^{G12D} LUAD with the data in prior reports of other mouse cancer models, so as to include and highlight the mentioned work by You et al regarding KRAS^{G12D}-induced myeloproliferative neoplasms (new reference #17 added) while also citing our own work regarding the role of SOS1 in DMBA/TPA-induced Ras skin carcinogenesis and in Bcr/Abl-induced CML (references #32-34).

2. *The sentence "The quest for new therapies must also take into account the experimental evidence demonstrating that the activity of non-mutated (WT) cellular RAS proteins is critically required for the development of oncogenic RAS-driven tumors..." does not entirely capture the complexity of wild-type RAS contributions to cancer. On one hand, Zang et al., demonstrated in a Nature Genetics paper published in 2001 that Kras^{+/-} mice are more susceptible to urethane carcinogenesis while Lim et al., demonstrated in a Nature paper published in 2008 that*

knockdown of wild-type RAS inhibited xenograft tumor growth of pancreatic cancer cell lines. One interpretation of these opposite results is that wild-type RAS proteins inhibit tumor initiation but promote later stages of tumorigenesis. Nevertheless, the role of wild-type RAS proteins in tumorigenesis is more complicated than described by the above the authors. Given that the authors study an activator of wild-type RAS, could they please provide a more comprehensive review of the literature, which will better set the stage for why their current work is important.

Reply. Again, we totally agree with this reviewer's comment. As for the previous point, the constraints regarding word count and number of references probably led us to oversimplify the complexity of the mechanistic contributions of WT RAS to cancer in this introductory statement of the originally submitted manuscript.

In our currently revised version, we have now modified and extended the content of this sentence as suggested by the reviewer, and have also added several new references, including those mentioned in her/his comments (new references #12 and #13), so as to provide a more comprehensive scope of the relevant literature (lines 55-60). Furthermore, we have also added a related new sentence in Discussion in order to further remark this point (lines 395-400).

3. *The authors describe, rather than state their quantification. For example, "Quantitation of the number of tumors visible on the pleural surface of the lungs of 1-5 month-old mice during these early months of tumor initiation and development in this mouse model showed that the total tumor count in SOS1KO/KRASG12D and SOS2KO/KRASG12D mice was dramatically reduced in comparison to their control SOS1/2WT/KRASG12D counterparts." Rather than using descriptors, could the authors please first report the actual quantification, then layer in their descriptors.*

Reply. Following the reviewer's recommendation, we have modified the description of these data in the currently revised version of the manuscript by first reporting the actual quantification of the numbers of pleural surface tumors observed in the lungs of the different genotypic groups of interest during these early stages of tumor initiation and development (lines 149-156).

4. *Please elaborate the separation of the two roles of Sos1. Significant decrease in CAFs in tumor microenvironment was observed when Sos1-KO mice were injected with KPB6 cells (Figure 4) but not when WT mice were injected with Sos1/2-WT, Sos1-KO, or Sos2-KO KPB6 cells (Figure 5). This suggests that the role of Sos1 in altering the tumor microenvironment in the recipient mice is separate from its impact on the growth of tumor initiating cell.*

Reply. Agree. As the reviewer aptly notes, the effect of SOS1 ablation on the TME CAF population appears to be separate from its impact on the tumoral cell itself when comparing the data in Fig 5 and Fig 6. Following the reviewer's recommendation to elaborate on separation of roles of SOS1, we have added a new statement in this regard to the Discussion (lines 410-416) of our current revised version of the manuscript.

5. *Please clarify what tissues Cre excision of Sos1 and Sos2 occurs in, and how this ablation of the genes was confirmed.*

Reply. As reported in our previous publications, our studies of SOS1/2 knockout strains usually involve whole-body excision of these GEFs (SOS1 excision induced by TMX in the diet, and constitutive SOS2 disruption; references #25, #26), and we routinely test for SOS1/2 ablation in different cells or tissues (thymus, skin, etc) by means of PCR or WB assays (references #26, #27, #32 and #42). To address this reviewer's comment, we are now adding two new panels and corresponding legends in Figure 2 of our currently revised manuscript to confirm SOS1/2 ablation (Fig. 2C: RT-qPCR and Western blot) in lung tumor tissue of the relevant genotypes. Relevant information concerning methodology is included in the corresponding Figure legend and M&M section.

6. The authors state that a “critical requirement of SOS1 for in vivo initiation, maintenance, and progression of KrasG12D-driven LUAD in mice” but this is supported by a single genetic background and a single tested LUAD murine cell line. Perhaps the authors could state what they demonstrate, then speculate on the wider possibilities of their findings in regard to lung cancer in general. This is somewhat of a reoccurring theme, for example, the experimental studies are all performed in mice, so perhaps noting this caveat when speculating on the role of SOS proteins in human cancer is warranted.

Reply. We acknowledge and thank this reviewer cautionary comment regarding potential overreaching with the conclusions and implications of this work, whose scope is certainly limited to our specific model of mouse KRAS^{G12D}-LUAD. As noted by the reviewer, we always paid attention to end our sentences with “in mice” in the various statements distributed throughout our original manuscript (“critical requirement of SOS1 for in vivo initiation, maintenance, and progression of KrasG12D-driven LUAD in mice”) but we certainly recognize the opportuneness to further highlight and address this possible caveat in the currently revised version, so as to avoid any possible overreaching when discussing the role of SOS proteins in human cancer.

To address the limitation of having used a single murine LUAD cell line, in the current revision of our manuscript we are including additional data corresponding to two new, independent, KRAS^{G12} LUAD murine cell lines (LKR10 and LKR13; new panels in Fig 7D,E; described in lines 293-300 in Results section). Furthermore, consistent data recently reported for various human cell lines is also now mentioned in the Discussion (lines 395-400).

This point is also pertinent when discussing the *in silico* analyses of data from human tumor databases and their possible significance regarding the role of SOS proteins in human cancer. In this regard, in our currently revised manuscript, we have modified the Discussion paragraph dealing with those database analyses so as to highlight the notion that our work was done in mice (line 428-431) while still referencing various literature reports describing consistent observations in human clinical samples (references #57 and #58) (lines 437-438).

7. The authors introduce targeting SOS1 as a therapeutic strategy partially due to its important role in activation of the WT KRAS, but there was no further exploration of the impact of genetic ablation of Sos1 on the WT Kras activity or signaling. Please either measure WT Kras/Nras/Hras GTP loading in the tumor setting or indicate the caveat to this conclusion is that the level of wild-type RAS activation was not determined as SOS can activate other small GTPases.

Reply. We totally agree and recognize this important point mentioned by the reviewer. Indeed, our current data in this report does not provide a direct experimental measurement of the effect of SOS1 ablation on WT RAS activity/signaling, and therefore only suggests that the therapeutic effect of SOS1

ablation in KRAS^{G12D}-driven LUAD may be due, at least partially, to its impact on activation of WT RAS proteins.

Unfortunately, in our hands, and with currently commercially available RAS antibodies, we have not been able to specifically differentiate among the various canonical RAS forms, thus impeding the possibility to precisely measure and distinguish among WT KRas/NRas/HRas GTP levels after SOS1 or SOS2 ablation in our KRAS^{G12D/+} LUAD mouse model. On the other hand, as a variety of reports from our own lab (references #28, #32) and other laboratories (references #12, #13, #16, #17, #53 and #54) also support the notion of significant WT involvement in RAS^{mut} oncogenesis, we have now added a new sentence and related literature references to the Discussion of our revised manuscript (lines 395-400) in order to more thoroughly discuss these important issues.

8. The in silico data in the final figure does not suggest dependency of LUAD cell lines on SOS1, but rather a slightly higher dependency on SOS1 than SOS2. In DepMap, the dependency of LUAD cells on KRAS or SHP2 is much higher than SOS1, but another RAS-GEF RASGRP2 is akin to SOS1. Please consider modifying the text to clarify the in silico analysis.

Reply. We agree with the reviewer's indication of the need to clarify the actual relevance of the DepMap dependency scores in our report. In fact, consistent with her/his assessment, our description in the Results section of our originally submitted manuscript was specifically limited to saying only that our analyses of LUAD cell lines "assigned a higher dependency score to hSOS1 in comparison to hSOS2, which showed an almost null dependency score...." (line 312 of Results; line 425 Discussion section). On the other hand, to further clarify our *in silico* analyses and the relative relevance of the SOS1 dependency scores in comparison to other tumor types or genes, we have now modified the Discussion in our revised version of the manuscript by adding a new sentence comparing the SOS1 scores in LUAD to those reported in other tumor types like CML ("Of note, development"; lines 428-431).

9. The authors previously demonstrated that Sos1, but not Sos2, was critical for early thymocyte maturation in their Molecular & Cellular Biology paper from 2013. Here, they beautifully show a decrease in CAFs, TAMs, and T-lymphocytes in TME with Sos1, but not Sos2, ablation. Could the authors please provide data demonstrating that TME changes were not due to the hematopoietic composition changes in the whole animal and, rather, are specific to the TME?

Reply. Regarding this issue it is relevant to point out that, although our previous work demonstrated that Sos1, but not Sos2, was critical for early thymocyte maturation, our detailed evaluation of various specific hematological parameters in the peripheral blood of our KO animals demonstrated that the single SOS1^{KO} or SOS2^{KO} mice did not show any significant alteration of those parameters in comparison to the SOS1/2^{WT} normal controls, and only the double knockout SOS1^{KO}/SOS2^{KO} mice displayed significant alterations of hematopoietic composition in our studies (reference #26).

In particular, regarding the T lymphocyte population, we showed that the cellular expansion taking place during the thymocyte maturation process was only affected when both SOS1 and SOS2 were concomitantly absent (but not in single KO SOS1^{KO} or SOS2^{KO}) and we also showed that only the Sos1/2 DKO mice displayed a marked decrease in the absolute number of T cells (both in peripheral blood and the spleen) whereas single SOS1^{KO} or SOS2^{KO} disruption did not alter the mature T-cell count in those tissues. Regarding the macrophage population, our prior studies have also shown that single SOS1 depletion did not affect the percentage of monocytes in the peripheral blood (reference #26). Interestingly, a previous report (reference #41) describing SOS1 regulation of podosome assembly and invasive capacity in macrophages suggests that macrophage activation, rather than macrophage count and viability might be altered in the single SOS1^{KO} animals.

To address this particular point, a new sentence has now been added in the Discussion of our currently revised version of the manuscript in order to further clarify this issue (“*These TME alterations WT mice*”, lines 372-376).

10. *Replicating the growth impairment phenotype with genetic ablation and pharmacological inhibition in Figure 7 with another KrasG12D-mutant LUAD cell line would help strengthen the claims regarding a KrasG12D-mutant LUAD-specific SOS1 dependency.*

Reply. Following the reviewer’s suggestion, and based on our recent work during the past two months, we have been able to add new data related to two more murine KRAS^{G12D}-mutated LUAD cell lines (LKR10 and LKR13; reference #43) and one more specific KRAS^{G12D} inhibitor (MRTX1133; new reference #68) to the data set initially presented in Fig 7 of our originally submitted manuscript. The new data is now being added as new figure panels 7D and 7E in the revised version of our paper.

Consistent with the previous data, these new results showed that BAY293-mediated interruption of SOS1:KRAS interaction resulted in a significant reduction of cell growth in the LKR10 and LKR13 cell lines, as previously observed in the KPB6 cell line (Fig 7D). These results indicated that BAY293 produces similar (or may be slightly lower) overall inhibitory effect than the specific KRAS^{G12D} inhibitor MRTX1133. We have also examined the effect of single or combined treatment with BAY293 and MRTX1133 on the activation of RAS upon EGF stimulation in the three murine LUAD cell lines (Fig 7E). Interestingly, in these particular cell lines, our Ras-GTP measurements did not seem to show any synergistic inhibitory effect after combined BAY293 + MRTX1133 treatment. All these new data are now described in a short sentence added to the corresponding section of Results (lines 293-300) in the revised version of our paper.

Regarding this point, it is also relevant to mention the paragraph of Discussion mentioning a recent paper by Hoffman *et al* that, consistent with our results, describes other different human LUAD cell lines holding KRAS^{G12D} mutations which also exhibit SOS1 dependency as demonstrated by assays using another specific (BI3406) SOS1 inhibitor (reference #55; line 450 of the revised manuscript).

We have also tried to confirm the growth impairment phenotype upon SOS genetic depletion (CRISPR/Cas9-mediated) in the new LKR10 and LKR13 LUAD cell lines. Unfortunately, because of various technical difficulties, in this case we have not been able to successfully complete the experiments on time before resubmission of our currently revised manuscript, and therefore those experiments with these or other cell lines will need to be completed in future.

REVIEWER #2 (Remarks to the Author):

In this work, Baltanás and colleagues describe the generation of KRAS mutant/SOS1-2 KO genetic engineered mouse model of lung cancer.

KRAS targeting has become particularly relevant since the approval of KRAS G12C covalent inhibitors in clinic and SOS1 inhibitors are in clinical testing meaning that the development of these in vivo models is very valuable.

I would point out to the authors that the concept of constitutively activation of mutant KRAS is bypassed by the more recent data showing that KRAS mutant alleles are subjected to GEF-mediated activation, GAP-mediated inactivation, and interaction with downstream effectors in different ways (for example PMID: 23487764 and PMID: 26037647 and other works on KRAS G12C inhibitors-based combinations), and the contribution of RAS wt isoforms is not the main or the unique source of RAS pathway re-activation upon RAS inhibition.

Having said that, I appreciated the description provided in this manuscript both at the tumor intrinsic and tumor extrinsic levels.

The mainly cytostatic effect of SOS1 KO (and to a lesser extent, SOS2 KO) is interesting and suggests that SOS1 inhibition should be combined with other inhibitors to successfully induce cell death, as previously reported. This makes these GEMM models very useful on the long term to study acquired resistance.

Reply. We acknowledge and appreciate this reviewer's assessment of the significance of our work in the context of the current state of understanding in the field regarding the recent approval for clinical testing of different, specific KRAS^{mut} and SOS1 inhibitors, as well as the emerging body of experimental evidence uncovering a wide range of varying signaling intensities resulting from functional /productive interactions between different, specific RAS^{mut} proteins with upstream GEF and/or GAP regulators. In the current revised version of our manuscript, we are trying to better convey this more comprehensive, updated view of the field by modifying and adding a new sentence and literature references mentioned by the reviewer to the text contained in the first paragraph of Introduction (lines 55-60, new references #18 and #19 in the revised manuscript).

We also recognize the reviewer's stated appreciation of "the description provided in this manuscript at the tumor intrinsic and tumor extrinsic levels". We thank the reviewer for this statement including a very clarifying terminology (tumor intrinsic, tumor extrinsic) that we have now incorporated at different places in the currently revised version of the Discussion in our manuscript (i.e., lines 378-380, 403 or 416).

Regarding the reviewer's remarks about the mainly cytostatic effect of SOS1 both *in vivo* (mouse tumor model, Figs 1-4) and *in vitro* (cell lines, Figs 5-7) and the convenience of combining SOS inhibition with other inhibitors, in our currently revised manuscript, we have been able to incorporate some recent, additional data about individual or combined drug treatments (new panels 7C, D) involving an additional, specific KRAS^{G12D} inhibitor (MRTX1133) (reference #68) and two other LUAD cell lines that were not yet available for testing in our laboratory at the time of our originally submitted manuscript. These new assays are now described in lines 293-300 of the Results section in the revised manuscript. Consistent with the reviewer's remarks, we also plan to carry out in the near future more extensive *in vivo* tests/assays of these and other additional, relevant drug combinations using our *in vivo* mouse KO tumor models.

-- I do not particularly like the human validation as final part of the work. In my opinion, these kinds of analyses should be presented at the very beginning to further highlight the importance of SOS1 role in the RAS-targeting and not to show that human data replicate what happen in mice. I would suggest moving these data at the beginning, merged with figure 1. Also figure 8D is missing the SOS1/SOS2 labeling.

Reply. We thank the reviewer for noticing the missing labeling in Fig 8D. Proper SOS1 and SOS2 labels are now included in the corrected Fig 8 included in the revised version of our manuscript.

Regarding her/his comments on our *in silico* analyses of human LUAD data, we recognize that our reported SOS1 dependency scores do not provide any direct demonstration that this GEF is essential for human LUAD, and only indicate that the dependency of human LUAD cell lines is higher for SOS1 than for SOS2. Also, the analyses correlating SOS1 expression with human LUAD development and patient survival just constitute circumstantial observations that are not inconsistent with the conclusions drawn from our experimental work in our mouse models. Based on all these considerations underscoring the merely circumstantial value of these *in silico* human data (see accordingly modified paragraph in our Discussion of the revised version, lines 428-431) and our response to related comments from the other reviewers, we prefer to keep the current location of this section at the end of Results in the manuscript, as a way to somehow underscore the merely supporting, ancillary value of these data in contrast to the direct proving nature of the experimental work previously described in the manuscript using our *in vivo* and *in vitro* mouse LUAD models.

-- Regarding the tumors that develop in the SOS1 KO model, did authors conducted further analyses? Is RAS pathway active? Is it mediated by SOS2? If these tumors can be grown *in vitro*, would SOS2 silencing cause cell death? Or direct RAS inhibition would work better?

Reply. Regarding this comment, it is relevant to mention that the primary experimental observations described in this report about the effect of SOS1 ablation in our mouse LUAD models were only produced very recently, and we have not had time to meaningfully address many important mechanistic questions like these mentioned here by the reviewer. In this regard, during the past 2-3 months we have only been able to satisfactorily complete some additional analyses that have been now included, as new figure data panels, in the revised version of our paper.

In particular, with regards to further analyses, the new panels in Fig 2C,D contain SOS1/2 expression assays and Ras.GTP pull-down assays performed on lung tumors of the 3 relevant genotypes (SOS1/2^{WT}, SOS1^{KO} and SOS2^{KO}), indicating that both SOS1 or SOS2 ablation significantly reduced RAS.GTP levels in the tumors. Furthermore, in the new panels added in Fig 7D,E, we have also evaluated the effect on cell proliferation and RAS activation of a specific inhibitor of KRAS:SOS1 interaction (BAY293) and a specific inhibitor of KRAS^{G12D} (MRTX1133) after adding them, individually or in combination, to cultures of three independent murine LUAD cell lines.

In any event, it is clearly apparent that much additional, extensive experimental work will have to be completed in the future in a comprehensive manner to meaningfully address many relevant mechanistic questions such as those mentioned in this reviewer's comment. In this regard, as suggested by the reviewer, and based on our previous characterization of the proliferative and apoptotic features of SOS1/2-DKO MEFs and the *in vivo* behavior of our tumor mouse models after concomitant disruption of SOS1 and SOS2 (references #26, #27, #28, #30, #32) we are keenly very interested to find out in future whether the SOS1^{KO} tumors can be grown *in vitro*, or if additional SOS2 silencing may increase cell death rates in those tumors.

-- Given the relevance of the journal and topic, I think some *in vivo* experiments using pharmacological treatments would be beneficial. I would be curious to know if concomitant G12D inhibition would synergize with SOS1 ablation and/or if SOS1 inhibition would synergize with SOS2 KO. As for now, I believe that this is the biggest lack in this story.

Reply. We certainly agree with the reviewer's statement emphasizing the appropriateness of the *in vivo* experiments mentioned in this query. However, as already mentioned above, we also realize that only after completing in the future and extensive amount of additional research work, we will be able to obtain

definite, clear mechanistic answers drawn from performing comprehensive *in vivo* experiments involving *in vivo* pharmacological treatments with individual drugs or combinations thereof.

For example, so far we have only been able to tackle very partially the question of whether concomitant G12D ablation would synergize with SOS1 and/or SOS2 ablation. Given that genetic DKO SOS1/2 mice are not viable (reference #26), this type of assays would need to be done using available pharmacological inhibitors against KRAS or SOS. Unfortunately, the only SOS1 inhibitor available to us at the time of performing the experiments described in this report (BAY293) was not orally bioavailable, thus precluding the performance of meaningful, long-term *in vivo* assays using our mouse model. Indeed, because of the limited availability of drugs, and the time constraint to send our current response to reviewers comments, at present we are only able to present the new data in panels Fig 7D,E of our currently revised manuscript. In this work we have evaluated this potential synergistic effect of G12D inhibition (by using the specific RAS^{G12D} inhibitor MRTX1133) and SOS1 ablation (BAY-293 mediated) in three different KRAS^{G12D}-mutated murine LUAD cell lines (KPB6, LKR10 and LKR13). Our results demonstrated an antiproliferative effect of both compounds when they are individually administered (Fig 7D), although it appears that MRTX1133 generally exhibits a more potent effect that seemed not to be further potentiated when both inhibitors were concomitantly administered.

For the future, our immediate plans include the acquisition of other commercially available, specific SOS1 inhibitors that are orally bioavailable so as to make it possible *in vivo* analysis of individual or combined pharmacological treatments in our LUAD mouse model.

REVIEWER #3 (Remarks to the Author):

The manuscript by Baltanas et al uses genetic approaches to assess the independent roles of SOS1 and SOS2 in KRASG12D-driven lung adenocarcinoma (using the Tyler Jacks LA2 strain that gives rise to spontaneous LUAD). The authors show that each RASGEF plays a role in tumorigenesis, however, the effect of Sos1 KO was much more prominent than for Sos2 KO. For SOS2, the authors show a role in tumor initiation: Young Sos2 KO mice (1-5 months) show a lower number of surface tumors and a lower overall tumor burden compared to WT controls, but these differences become less pronounced as mice age so that the overall effect on survival is not significant (although the differences in survival at < 200 days look convincing). This effect on tumor initiation is even more pronounced in adoptive transfer experiments where SOS2 was deleted using CRISPR (See Fig. 6), where the effects were more similar to SOS1 KO.

For SOS1, the overall effects on tumorigenesis were more pronounced. The authors use Sos1^{fl/f} mice where the floxed allele is not removed until 1 month of age and show a more pronounced early and late reduction in tumor burden compared to Sos2 KO, along with a significant enhancement in overall survival. Sos1 KO in older mice (4 months) also caused partial regression of established tumors, showing that SOS1 is a potential therapeutic target. The authors further show that these effects effect of SOS1 ablation on tumorigenesis were due both reduced tumor cell proliferation (reduced Ki67 and pERK staining) and on the tumor microenvironment (reduced SMA staining for cancer-associated fibroblasts). The authors confirm these two roles for SOS1 in adoptive transfer experiments: transfer of Sos^{WT} tumor cells into Sos1 KO recipient mice showed reduced tumor burden that was due to low numbers of CAFs (tumor extrinsic); transfer of Sos1 KO cells into WT recipient mice showed reduced tumor burden (tumor intrinsic). In contrast, SOS2 only showed tumor intrinsic effects. Finally, the authors probed both DepMap and CANCEPTOOL databases to show a correlation between SOS1 expression and cell line (DepMap) or patient (CANCEPTOOL) survival.

This is a fantastic paper, and the authors should be commended for their tremendous work. The experiments are well thought out, and the conclusions are all supported by the data. While I have several specific suggestions that will help bolster the authors' arguments, I do not think any additional experimentation is needed (well done by all).

Reply. We are grateful and flattered by this reviewer's remarkably positive evaluation of our manuscript and her/his statements indicating that no additional experimentation is needed and that the authors should be commended for their work. Our answers to her/his specific suggestions to help bolster the authors' arguments are provided below.

Comments

I. *In the tumor regression experiments (Fig. 4), it is unclear as the data are currently presented whether the reduced tumor burden is due to true regression or only in stasis (inhibition of further proliferation). The authors present tumor volumes at 6 months...however they should (per the methods) have paired 4-month measurements for each mouse. For this data it is important for the authors to show these paired measurements -preferably for each mouse. This would be a paired statistic showing 4 and 6 month measurements in the Sos1^{fl/f} mice and the Sos1^{KO} mice. For an example in an unrelated field see Lake et al, Cell Death and Disease 12:400 (Fig. 1B-D). This is especially important given the lack of differences in their cell death (CC3) staining.*

Reply. We agree with the reviewer's statement and apologize if our representation of results in Figure 4A of the original version of our manuscript was not informative enough or could lead to some confusion. According to the reviewer's advice, we have now identified the data points corresponding to each individual mouse with separate, distinct colors in the revised version of panel 4A and also performed paired t-tests for statistical analysis that revealed significant differences between the groups ($p = 0.0173$). This new information has now been added to the corresponding figure legend (lines 717-718) and the M&M section dealing with statistics (lines 591-592).

2. *In the discussion (lines 376-379) the authors state ‘it remains to be determined whether ...the effect of SOS1 ablation is due to loss of activation of the resident KRASG12D mutant proteins or the non-mutated KRASWT proteins. I submit that if the effects of SOS1 ablation is on WT proteins, it is very likely that inhibition of WT HRAS and WT NRAS in both the tumor and the stroma are significantly important as well. There is substantial evidence for the role of WT RAS family members, distinct from the mutant allele, in mutant KRAS-driven tumorigenesis. This should be expanded on in the discussion. This comes up again on line 422 – where SOS1 inhibitors also block WT HRAS and NRAS.*

Reply. We certainly agree with this reviewer’s views regarding the convenience to expand our discussion about the potential impact of SOS1 ablation on activation of RAS^{WT} proteins in tumor and stroma, as well as the role of WT RAS family members, distinct from the mutant allele, in mutant KRAS-driven tumorigenesis.

Following her/his request to expand discussion on these issues, and in line with our response to related comments from another reviewer, we have now modified and added new literature references to a sentence in Introduction (lines 55-60), and have also added a complete new sentence and relevant literature references in the Discussion (lines 395-400) so as to provide a more comprehensive, thorough discussion of these issues in the currently revised version of our paper.

3. *On lines 381-386, the adoptive transfer into Sos1KO recipients argues for a non-cell autonomous function...the authors should be careful to clarify the discussion here.*

Reply. Regarding this reviewer comment, we think that talking about tumor intrinsic or extrinsic effects of SOS1 ablation is a much better terminology than “...to test the cell-autonomous nature---” to refer to the issues discussed in this particular sentence, and therefore we have modified this text accordingly in this Discussion section of the revised version of our paper (lines 403-404 in the revised manuscript)

4. *In discussing the adoptive transfer of Sos1 KO versus Sos2 KO cells (lines 390-392), the authors hypothesize that the reduced tumor burden in Sos2 KO was due to altered cell homing to the heart. Is there data supporting this? Alternatively, this could simply reflect the role of SOS2 in tumor initiation but not in the TME.*

Reply. In our recent repetitions of these experiments, we have not detected any significant heart homing of SOS2^{KO} cells and therefore we are modifying our initial view (hypothesized in the mentioned statement of our original discussion) along the lines of the alternative suggested in this reviewer’s comment. In this regard, we have substituted the mentioned original sentence by a new, modified one included now in lines 410-416 of the revised version of our manuscript.

5. *The authors seem to miss an opportunity to really hammer the point that SOS1 effects are in both the tumor and in CAFs. Indeed, the authors previously showed a prominent role for SOS1 in proliferation of MEFs...these data support the CAF data.*

Reply. We recognize that this point was not sufficiently emphasized in our original version of Discussion and therefore we are heeding the reviewer’s advice to more strongly hammer in our revised version the notion that our data strongly indicate that SOS1 ablation in our KRAS^{G12D}.LUAD system impacts both the homeostasis/proliferation of the intrinsic tumoral cell population and also that of the extrinsic TME cell populations such as the CAFs mentioned here by the reviewer. In this regard, it is also relevant to mention that our *in vivo* data in this KRAS^{G12D} LUAD model are also consistent with previous work demonstrating

the prominent role of SOS regulating the proliferation and homeostasis of primary MEF fibroblasts (references #27 and #28) as well as the responsiveness and migratory ability of primed neutrophils and macrophage populations in inflammation or wound healing (references #32, #41 and #42). These views are now summarized in a new, short sentence that is now inserted in one paragraph of Discussion in the currently revised version of our manuscript (lines 378-380).

6. In the first paragraph of the discussion, the authors have repeated “uncovered also a measurable delaying effect of SOS2 ablation regarding the onset and initial stages of LUAD development in KRASG12D mice” on lines 325-327 and 328-330.

Reply. We thank the reviewer for noticing this error. The text duplication has been corrected in the currently revised version of the manuscript.

7. In Fig. 2B, it appears that Sos2 KO also showed reduced pERK relative to WT. Is this significant?

Reply. Although, as noted by the reviewer, it appears that pERK levels, in SOS2^{KO} tumors seem to be reduced in comparison to SOS1/2^{WT} counterparts, our statistical analysis, however, did not yield statistically significant differences (p=0.0597) between both experimental groups.

REVIEWERS' COMMENTS

Reviewer #1 (Remarks to the Author):

all my concerns have been addressed.

Reviewer #2 (Remarks to the Author):

The Authors nicely addressed all of my questions, I am looking forward to reading the published paper and all the follow up studies that will come after this one.

Reviewer #3 (Remarks to the Author):

The manuscript by Baltanas et al is a revision of an earlier submission that uses genetic approaches to assess the independent roles of SOS1 and SOS2 in KRASG12D-driven lung adenocarcinoma.

Significant to the revision, the authors have:

- Performed paired T tests on data from each mouse in the tumor regression experiments presented in Fig. 4.
- Performed new experiments assessing combined G12Di (MRTX1133) and SOS1i (BAY-293) in Fig 7.
- Made significant changes to the text based on comments from all three reviewers.

This is a great manuscript, and will move the field forward. All of my concerns have been addressed , and I look forward to seeing the published manuscript.

Reviewer #4 (Replacement reviewer for Reviewer #1, Remarks to the Author):

In this manuscript, Baltanas et al compare SOS1 versus SOS2 dependency in murine Kras G12D driven lung adenocarcinoma, and identify a key role for SOS1 in lung tumor development and maintenance. They also demonstrate impact on the TME, including depletion of CAFs and collagen, as well as depleted TAMs and T cells. They further validate the impact of SOS1 inhibition in murine Kras mutant lung adenocarcinoma cell lines, and perform several in silico analyses from human lung adenocarcinoma genomic databases suggesting preferential dependency on SOS1.

Overall this is an outstanding manuscript, and the reviewers have satisfactorily addressed Reviewer #1's comments.

My only additional comment with respect to the TME studies is that they are being conducted at endpoint, so may not reflect the acute consequences of SOS1 inhibition. This can be addressed via the text especially in the discussion - that future studies of acute SOS1 pharmacologic inhibition in the mouse would enable studies of changes in the lung cancer TME over time. My suspicion is that actively dying tumors would recruit an inflammatory infiltrate. And what they are observing are the sequelae of chronic inflammation.

REVIEWER #4 (Remarks to the Author):

In this manuscript, Baltanas et al compare SOS1 versus SOS2 dependency in murine Kras G12D driven lung adenocarcinoma, and identify a key role for SOS1 in lung tumor development and maintenance. They also demonstrate impact on the TME, including depletion of CAFs and collagen, as well as depleted TAMs and T cells. They further validate the impact of SOS1 inhibition in murine Kras mutant lung adenocarcinoma cell lines, and perform several in silico analyses from human lung adenocarcinoma genomic databases suggesting preferential dependency on SOS1.

Overall this is an outstanding manuscript, and the reviewers have satisfactorily addressed Reviewer #1's comments.

My only additional comment with respect to the TME studies is that they are being conducted at endpoint, so may not reflect the acute consequences of SOS1 inhibition. This can be addressed via the text especially in the discussion - that future studies of acute SOS1 pharmacologic inhibition in the mouse would enable studies of changes in the lung cancer TME over time. My suspicion is that actively dying tumors would recruit an inflammatory infiltrate. And what they are observing are the sequelae of chronic inflammation.

Reply. We gratefully acknowledge this reviewer's high enthusiasm for our study, as well as her/his overall really positive assessment of the work and conclusions included in the manuscript. We agree with reviewer's assessment about TME response may not reflect the acute consequences of SOS1 inhibition. Following referee's recommendation we have included a sentence in the Discussion section in order to further remark this point (lines 454-455).